# Energy-Based Cross Attention for Bayesian Context Update in Text-to-Image Diffusion Models

**Geon Yeong Park**[1*] **Jeongsol Kim**[1*] **Beomsu Kim**[2] **Sang Wan Lee**[1,2,3†] **Jong Chul Ye**[2†]

[1]Bio and Brain Engineering, [2]Kim Jaechul Graduate School of AI, [3]Brain and Cognitive Sciences

Korea Advanced Institute of Science and Technology (KAIST), Daejeon, Korea

*, †: Co-first and Co-corresponding authors

{pky3436, jeongsol, beomsu.kim, sangwan, jong.ye}@kaist.ac.kr

## Abstract

Despite the remarkable performance of text-to-image diffusion models in image generation tasks, recent studies have raised the issue that generated images sometimes cannot capture the intended semantic contents of the text prompts, which phenomenon is often called semantic misalignment. To address this, here we present a novel energy-based model (EBM) framework for adaptive context control by modeling the posterior of context vectors. Specifically, we first formulate EBMs of latent image representations and text embeddings in each cross-attention layer of the denoising autoencoder. Then, we obtain the gradient of the log posterior of context vectors, which can be updated and transferred to the subsequent cross-attention layer, thereby implicitly minimizing a nested hierarchy of energy functions. Our latent EBMs further allow zero-shot compositional generation as a linear combination of cross-attention outputs from different contexts. Using extensive experiments, we demonstrate that the proposed method is highly effective in handling various image generation tasks, including multi-concept generation, text-guided image inpainting, and real and synthetic image editing. Code: `https://github.com/EnergyAttention/Energy-Based-CrossAttention`.

## 1 Introduction

Diffusion models (DMs) have made significant advances in controllable multi-modal generation tasks [37], including text-to-image synthesis. The recent success of text-to-image diffusion models, e.g., Stable Diffusion [31], Imagen [33], etc., is mainly attributed to the combination of high-fidelity DMs with high-performance large language models (LMs).

Although text-to-image DMs have shown revolutionary progress, recent studies have shown that the current state-of-the-art models often suffer from semantic misalignment problems, where the generated images do not accurately represent the intended semantic contents of the text prompts. For example, [4] discovered the catastrophic neglect problem, where one or more of the concepts of the prompt are neglected in generated images. Moreover, for a multi-modal inpainting task with text and mask guidance, [41] found that the text-to-image DMs may often fail to fill in the masked region precisely following the text prompt.

Therefore, this work focuses on obtaining a harmonized pair of latent image representations and text embeddings, i.e., context vectors, to generate semantically aligned images. In order to mitigate the misalignment, instead of leveraging fixed context vectors, we aim to establish an *adaptive* context by modeling the posterior of the context, i.e. $p(\text{context} \mid \text{representations})$. Note that this is a significant departure from the previous methods which only model $p(\text{representations} \mid \text{context})$ with *frozen* context vectors encoded by the pretrained textual encoder.

37th Conference on Neural Information Processing Systems (NeurIPS 2023).

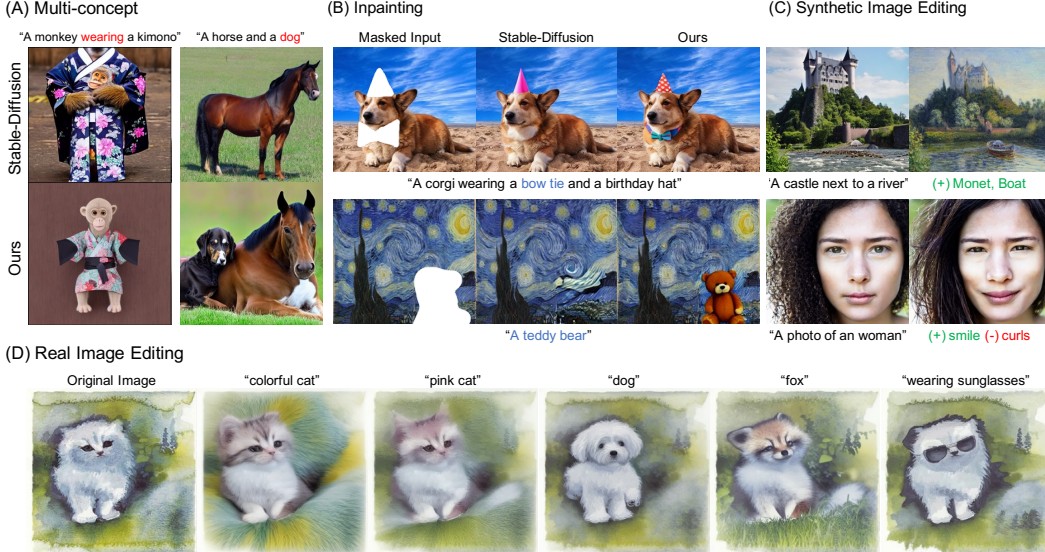

Figure 1: The energy-based cross-attention improves the semantic alignment between given text and the generated sample. The proposed method could be leveraged for multiple applications without additional training.

Specifically, we introduce a novel energy-based Bayesian framework, namely energy-based cross-attention (EBCA), which approximates maximum a posteriori probability (MAP) estimates of context vectors given observed latent representations. Specifically, to model $p(\text{context} \mid \text{representations})$, we first consider analogous $p(K_l|Q_{t,l})$ in the latent space of the intermediate cross-attention layer of a denoising auto-encoder, i.e., cross-attention space. Here, $K_l$ and $Q_{t,l}$ correspond to the key and query in $l$-th layer at time $t$ that encode the context from the text and the image representation, respectively. Inspired by the energy-based perspective of attention mechanism [30], we then formulate energy functions $E(K_l; Q_{t,l})$ in each cross-attention space to model the posterior. Finally, we create a correspondence between the context and representation by minimizing these parameterized energy functions. More specifically, by obtaining the gradient of the log posterior of context vectors, a nested hierarchy of energy functions can be implicitly minimized by cascading the updated contexts to the subsequent cross-attention layer (Figure 2) .

Moreover, our energy-based perspective of cross-attention also allows zero-shot compositional generation due to the inherent compositionality of Energy-Based Models (EBMs). This involves the convenient integration of multiple distributions, each defined by the energy function of a specific text embedding. In practical terms, this amalgamation can be executed as a straightforward linear combination of cross-attention outputs that correspond to all selected editing prompts.

We demonstrate the effectiveness of the proposed EBM framework in various text-to-image generative scenarios, including multi-concept generation, text-guided inpainting, and compositional generation. The proposed method is training-free, easy to implement, and can potentially be integrated into most of the existing text-to-image DMs.

## 2    Preliminaries

### 2.1    Diffusion Model

Diffusion models [36, 13, 37, 18, 16] aims to generate samples from the Gaussian noise by iterative denoising processes. Given clean data $\boldsymbol{x}_0 \sim p_{data}(\boldsymbol{x}_0)$, diffusion models define the forward sampling from $p(\boldsymbol{x}_t|\boldsymbol{x}_0)$ as $\boldsymbol{x}_t = \sqrt{\bar{\alpha}_t}\boldsymbol{x}_0 + \sqrt{1 - \bar{\alpha}_t}\boldsymbol{z}_t$, where $\boldsymbol{z}_t \sim \mathcal{N}(0, \boldsymbol{I})$, $t \in [0, 1]$. Here, for the Denoising Diffusion Probabilistic Model (DDPM) [13], the noise schedule $\beta_t$ is an increasing sequence of $t$, with $\bar{\alpha}_t := \prod_{i=1}^{t} \alpha_t$, $\alpha_t := 1 - \beta_t$. The goal of diffusion model training is to obtain a neural network $\boldsymbol{\epsilon}_{\theta^*}$ that satisfies

$$\theta^* = \underset{\theta}{\arg\min}\, \mathrm{E}_{\boldsymbol{x}_t \sim p(\boldsymbol{x}_t|\boldsymbol{x}_0), \boldsymbol{x}_0 \sim p_{data}(\boldsymbol{x}_0), \boldsymbol{\epsilon} \sim \mathcal{N}(0, \mathbf{I})} \left[ \|\boldsymbol{\epsilon}_\theta(\boldsymbol{x}_t, t) - \boldsymbol{\epsilon}\|_2^2 \right], \tag{1}$$

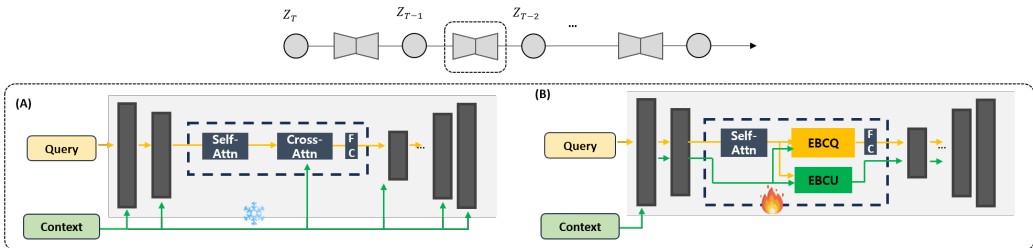

Figure 2: Comparison between the Stable-Diffusion and the proposed method. (**A**) The Stable-Diffusion uses fixed context embedding encoded by pre-trained CLIP. (**B**) The proposed method allows adaptive context embedding through energy-based context update (EBCU) and energy-based composition of queries (EBCQ).

so that the reverse sampling from $q(\boldsymbol{x}_{t-1}|\boldsymbol{x}_t, \boldsymbol{x}_{\theta^*}(\boldsymbol{x}_t, t))$ is achieved by

$$\boldsymbol{x}_{t-1} = \frac{1}{\sqrt{\alpha_t}}\left(\boldsymbol{x}_t - \frac{1-\alpha_t}{\sqrt{1-\bar{\alpha}_t}}\boldsymbol{\epsilon}_{\theta^*}(\boldsymbol{x}_t, t)\right) + \sigma_t\boldsymbol{z}_t, \tag{2}$$

where $\boldsymbol{z}_t \sim \mathcal{N}(0, \boldsymbol{I})$ and $\sigma^2$ is variance which is set as $\beta$ in DDPM. With iterative process, we can sample the image $\boldsymbol{x}_0$ from initial sample $\boldsymbol{x}_T \sim \mathcal{N}(0, \boldsymbol{I})$.

Since diffusion models require the iterative sampling on high dimensional space, they are computationally expansive and time consuming. To mitigate this limitation, Latent Diffusion Model (LDM) [31] has proposed diffusion processes on the compressed latent space using pre-trained auto-encoder. Furthermore, by introducing language model-based cross-attention to diffusion model's U-Net neural backbone [32], LDM enables token-based conditioning method such as text-to-image.

## 2.2 Energy-based Perspective of Attention Mechanism

**Definition.** Given an $N$-dimensional vector $\boldsymbol{v}$, its logsumexp and softmax are defined as

$$\text{logsumexp}(\boldsymbol{v}, \beta) \coloneqq \beta^{-1}\log(\textstyle\sum_{i=1}^N \exp(v_i)), \quad \text{softmax}(\boldsymbol{v}) \coloneqq \exp(\boldsymbol{v} - \text{logsumexp}(\boldsymbol{v}, 1)).$$

where $v_i$ denotes the $i$-th element of $\boldsymbol{v}$. For a given a matrix $\boldsymbol{A}$, its $\text{logsumexp}_i(\boldsymbol{A})$ and $\text{softmax}_i(\boldsymbol{A})$ are understood as taking the corresponding operation along the $i$-th dimension of $\boldsymbol{A}$. So, for instance, $i = 1$, $\text{softmax}_i(\boldsymbol{A})$ consist of column vectors that sum to 1.

**Energy model.** Following the success of the attention mechanism, several studies have focused on establishing its theoretical foundation. One promising avenue is interpreting the attention mechanism using the Energy-Based Model (EBM) framework. This line of research begins with recent research on modern Hopfield networks [30, 14], which gradually builds up to the self-attention mechanism of the Transformer model.

The Hopfield network is a dense associative memory model that aims to associate an input with its most similar pattern. Specifically, it constructs an energy function to model an energy landscape that contains basins of attraction around desired patterns. Recently, modern Hopfield networks [30, 6, 19] has introduced a new family of energy functions, which aim to improve its pattern storage capacity or make it compatible with continuous embeddings. To this end, [30] proposed the following energy function of a state pattern (query) $\boldsymbol{\zeta} \in \mathbb{R}^d$ parameterized by $N$-stored (key) patterns $\boldsymbol{X} = [\boldsymbol{x}_1 \ldots, \boldsymbol{x}_N] \in \mathbb{R}^{d \times N}$ and $\beta > 0$:

$$E(\boldsymbol{\zeta}; \boldsymbol{X}) = \frac{1}{2}\boldsymbol{\zeta}^T\boldsymbol{\zeta} - \text{logsumexp}(\boldsymbol{X}^T\boldsymbol{\zeta}, \beta). \tag{3}$$

Intuitively, the first term ensures the query remains finite, while the second term measures the individual alignment of the query with every stored pattern. Based on the Concave-Convex-Procedure (CCCP) [43], [30] derives the update rule for a state pattern $\mathcal{E}$ and time $t$ as follows:

**Proposition 1** ([30]). *Define the update rule* $f : \mathbb{R}^d \to \mathbb{R}^d$ *as follows:*

$$\boldsymbol{\zeta}^{new} = f(\boldsymbol{\zeta}) = \boldsymbol{X}\,\text{softmax}(\beta\boldsymbol{X}^T\boldsymbol{\zeta}) \tag{4}$$

*Then, the update rule decreases the loss (3) and converges globally. In another word, for* $\boldsymbol{\zeta}^{(t+1)} = f(\boldsymbol{\zeta}^{(t)})$, *the energy* $E(\boldsymbol{\zeta}^{(t)}) \to E(\boldsymbol{\zeta}^*)$ *for* $t \to \infty$ *and a fixed point* $\boldsymbol{\zeta}^*$.

Note that (4) is equivalent to a gradient descent update to minimize (3) with a step size $\eta = 1$:

$$\boldsymbol{\zeta} \leftarrow \boldsymbol{\zeta} - \eta \nabla_{\boldsymbol{\zeta}} \mathrm{E}(\boldsymbol{\zeta}; \boldsymbol{X}) = \boldsymbol{\zeta} - \eta(\boldsymbol{\zeta} - \boldsymbol{X} \,\mathrm{softmax}(\beta \boldsymbol{X}^T \boldsymbol{\zeta})). \tag{5}$$

**Connection to the attention of the transformer.** Remarkably, this implicit energy minimization is closely related to the attention mechanism as shown in [30]. To see this, suppose that $\boldsymbol{y}_i, \boldsymbol{r}_i \in \mathbb{R}^d$ is given as stored (key) and state (query) patterns, respectively. Let $\boldsymbol{W}_K, \boldsymbol{W}_Q \in \mathbb{R}^{d \times d_H}$ represent linear maps for $\boldsymbol{y}_i$ and $\boldsymbol{r}_i$, respectively. Then, we introduce $\boldsymbol{x}_i = \boldsymbol{W}_K^T \boldsymbol{y}_i \in \mathbb{R}^{d_H}$ and $\boldsymbol{\zeta}_i = \boldsymbol{W}_Q^T \boldsymbol{r}_i \in \mathbb{R}^{d_H}$. Let $\boldsymbol{Y} = (\boldsymbol{y}_1, \dots, \boldsymbol{y}_N)^T \in \mathbb{R}^{N \times d}$, and $\boldsymbol{R} = (\boldsymbol{r}_1, \dots, \boldsymbol{r}_S)^T \in \mathbb{R}^{S \times d}$. We define $\boldsymbol{X}^T = \boldsymbol{K} = \boldsymbol{Y}\boldsymbol{W}_K \in \mathbb{R}^{N \times d_H}$, and $\boldsymbol{Q} = \boldsymbol{R}\boldsymbol{W}_Q \in \mathbb{R}^{S \times d_H}$. By plugging $\boldsymbol{X}^T = \boldsymbol{K}$ and $\boldsymbol{\zeta} = \boldsymbol{q}_i$ into (4) for all $i$, we obtain that: $\boldsymbol{Q}^T = \boldsymbol{K}^T \,\mathrm{softmax}_1(\beta \boldsymbol{K} \boldsymbol{Q}^T) \in \mathbb{R}^{d_H \times S}$. By taking the transpose, we obtain $\boldsymbol{Q} = \mathrm{softmax}_2(\beta \boldsymbol{Q} \boldsymbol{K}^T)\boldsymbol{K}$. Let $\boldsymbol{V} \in \mathbb{R}^{N \times d_V}$ denote the value matrix as $\boldsymbol{V} = \boldsymbol{Y}\boldsymbol{W}_K\boldsymbol{W}_V = \boldsymbol{K}\boldsymbol{W}_V$, which will replace the $\boldsymbol{K}$ outside of $\mathrm{softmax}_2$. Then, we finally obtain

$$\boldsymbol{Q}^{new} = \mathrm{softmax}_2(\beta \boldsymbol{Q} \boldsymbol{K}^T)\boldsymbol{V}, \tag{6}$$

which is exactly the transformer attention with $\beta = 1/\sqrt{d_H}$. This connection affords us the insightful theoretical ground of the attention mechanism: the update step of the attention mechanism in a Transformer layer acts as an inner-loop optimization step, minimizing an implicit energy function that is determined by queries, keys, and values.

## 3 Energy-based Cross-Attention

Recall that our objective is to generate semantically correct images by harmonizing latent (U-Net) representations and context vectors within the denoising autoencoder. To achieve this, we propose a simple but effective Bayesian framework, namely energy-based cross-attention (EBCA). Specifically, we perform test-time optimization of context vectors within the cross-attention spaces of a denoising autoencoder to implicitly minimize a specially designed energy function. Note that this is a significant departure from the conventional text-guided diffusion models which have leveraged *fixed* context vectors embedded by pre-trained CLIP [29] to all cross-attention layers.

### 3.1 Energy-based Bayesian Context Update

**Energy function.** Our focus is on the cross-attention space of a time-dependent denoising auto-encoder, utilizing the conventional U-Net neural architecture. Here, we refer to latent representations as the representations of intermediate layers in U-Net unless otherwise specified. Let $L$ be the number of cross-attention layers. For each $l$-th layer at time $t$, we define the queries matrix $\boldsymbol{Q}_{t,l} \in \mathbb{R}^{P_l^2 \times d_l}$, and the keys and values matrices $\boldsymbol{K}_l \in \mathbb{R}^{N \times d_l}$ and $\boldsymbol{V}_l \in \mathbb{R}^{N \times d_l}$, respectively. Here, $P_l$ represents the spatial dimension of latent representations in the $l$-th layer. Given context vectors $\boldsymbol{C} \in \mathbb{R}^{N \times d_c}$, we map $\boldsymbol{K}_l$ and $\boldsymbol{V}_l$ with $\boldsymbol{W}_{K,l}, \boldsymbol{W}_{V,l} \in \mathbb{R}^{d_c \times d_l}$, such that $\boldsymbol{K}_l = \boldsymbol{C}\boldsymbol{W}_{K,l}$ and $\boldsymbol{V}_l = \boldsymbol{C}\boldsymbol{W}_{V,l}$. In the following, time $t$ and layer index $l$ are omitted in notations $\boldsymbol{Q}_{t,l}$ and $\boldsymbol{K}_l$ for simplicity.

The main goal is to obtain a maximum a posteriori probability (MAP) estimate of context vectors $\boldsymbol{C}$ given a set of observed latent representations of queries $\boldsymbol{Q}_{t,l}$. First, based on the energy functions (8) and (9), the posterior distribution of $\boldsymbol{K}$ can be defined by using Bayes' rule: $p(\boldsymbol{K} \mid \boldsymbol{Q}) = p(\boldsymbol{Q} \mid \boldsymbol{K})p(\boldsymbol{K})/p(\boldsymbol{Q})$, where (8) and (9) are leveraged to model the distribution $p(\boldsymbol{Q} \mid \boldsymbol{K})$ and $p(\boldsymbol{Q})$, respectively. Then, in order to obtain a MAP estimation of $\boldsymbol{C}$, we approximate the posterior inference using the gradient of the log posterior. The gradient can be estimated as follows:

$$\nabla_{\boldsymbol{K}} \log p(\boldsymbol{K} \mid \boldsymbol{Q}) = \nabla_{\boldsymbol{K}} \log p(\boldsymbol{Q} \mid \boldsymbol{K}) + \nabla_{\boldsymbol{K}} \log p(\boldsymbol{K}) = -\big(\nabla_{\boldsymbol{K}} \mathrm{E}(\boldsymbol{Q}; \boldsymbol{K}) + \nabla_{\boldsymbol{K}} \mathrm{E}(\boldsymbol{K})\big). \tag{7}$$

Motivated by the energy function in (3), we first introduce a new conditional energy function w.r.t $\boldsymbol{K}$ as follows:

$$\mathrm{E}(\boldsymbol{Q}; \boldsymbol{K}) = \frac{\alpha}{2} \mathrm{diag}(\boldsymbol{K}\boldsymbol{K}^T) - \sum_{i=1}^{N} \mathrm{logsumexp}(\boldsymbol{Q}\boldsymbol{k}_i^T, \beta), \tag{8}$$

where $\boldsymbol{k}_i$ denotes the $i$-th row vector of $\boldsymbol{K}$, $\alpha \geq 0$, and $\mathrm{diag}(\boldsymbol{A}) = \sum_{i=1}^{N} A_{i,i}$ for a given $\boldsymbol{A} \in \mathbb{R}^{N \times N}$. Intuitively, $\mathrm{logsumexp}(\boldsymbol{Q}\boldsymbol{k}_i^T, \beta)$ term takes a smooth maximum alignment between latent

representations $\boldsymbol{q}_j, j = 1, \cdots, P^2$ and a given text embedding $\boldsymbol{k}_i$. Let $j^* = \arg\max_j \boldsymbol{q}_j \boldsymbol{k}_i^T$. Then, the implicit minimization of $-\text{logsumexp}$ term encourages each $\boldsymbol{k}_i$ to be semantically well-aligned with a corresponding spatial token representation $\boldsymbol{q}_{j^*}$. In turn, this facilitates the retrieval and incorporation of semantic information encoded by $\boldsymbol{k}_i$ into the spatial region of $\boldsymbol{q}_{j^*}$.

The $\text{diag}(\boldsymbol{K}\boldsymbol{K}^T)$ term in (8) serves as a regularizer that constrains the energy of each context vector $\boldsymbol{k}_i$, preventing it from exploding during the maximization of $\text{logsumexp}(\boldsymbol{Q}\boldsymbol{k}_i^T, \beta)$, thereby ensuring that no single context vector excessively dominates the forward attention path. In this regard, we also propose a new $\boldsymbol{Q}$-independent prior energy function $\text{E}(\boldsymbol{K})$ given by:

$$\text{E}(\boldsymbol{K}) = \text{logsumexp}\left(\frac{1}{2}\text{diag}(\boldsymbol{K}\boldsymbol{K}^T), 1\right) = \log\sum_{i=1}^{N}\exp\left(\frac{1}{2}\boldsymbol{k}_i\boldsymbol{k}_i^T\right). \qquad (9)$$

Instead of penalizing each norm uniformly, it primarily regularizes the smooth maximum of $\|\boldsymbol{k}_i\|$ which improves the stability in implicit energy minimization. We empirically observed that the proposed prior energy $\text{E}(\boldsymbol{K})$ serves as a better regularizer compared to the original $\text{diag}(\boldsymbol{K}\boldsymbol{K}^T)$ term (related analysis in (12) and appendix E).

Although our energy function is built based on the energy function, in contrast to (3), the proposed one is explicitly formulated for implicit energy minimization w.r.t *keys* $\boldsymbol{K}$ and the associated *context vectors* $\boldsymbol{C}$, which is different from the theoretical settings in Section 2.2 w.r.t $\boldsymbol{Q}$ [30]. It is worth noting that the two energy functions are designed to serve different purposes and are orthogonal in their application.

**MAP estimation.** Based on the proposed energy functions, we derive the update rule of key $\boldsymbol{K}$ and context $\boldsymbol{C}$ following (7):

**Theorem 1.** *For the energy functions (8) and (9), the gradient of the log posterior is given by:*

$$\nabla_{\boldsymbol{K}}\log p(\boldsymbol{K} \mid \boldsymbol{Q}) = \text{softmax}_2\left(\beta\boldsymbol{K}\boldsymbol{Q}^T\right)\boldsymbol{Q} - \left(\alpha\boldsymbol{I} + \boldsymbol{\mathcal{D}}\left(\text{softmax}\left(\frac{1}{2}\text{diag}(\boldsymbol{K}\boldsymbol{K}^T)\right)\right)\right)\boldsymbol{K}, \qquad (10)$$

*where $\boldsymbol{\mathcal{D}}(\cdot)$ is a vector-to-diagonal-matrix operator.*

*Then, by using the chain rule the update rule of context vectors $\boldsymbol{C}$ is derived as follows:*

$$\boldsymbol{C}_{n+1} = \boldsymbol{C}_n + \gamma\left(\text{softmax}_2\left(\beta\boldsymbol{K}\boldsymbol{Q}^T\right)\boldsymbol{Q} - \left(\alpha\boldsymbol{I} + \boldsymbol{\mathcal{D}}\left(\text{softmax}\left(\frac{1}{2}\text{diag}(\boldsymbol{K}\boldsymbol{K}^T)\right)\right)\right)\boldsymbol{K}\right)\boldsymbol{W}_K^T, \qquad (11)$$

*where $\gamma > 0$ is a step size.*

In practice, we empirically observed that the nonzero $\alpha$ in (8) often leads to an over-penalization of contexts, which can ultimately cause some contexts to vanish. To address this, we set $\alpha = 0$. Moreover, we found it beneficial to assign distinct step sizes $\gamma_{\text{attn}}$ and $\gamma_{\text{reg}}$ as follows:

$$\boldsymbol{C}_{n+1} = \boldsymbol{C}_n + \gamma_{\text{attn}}\underbrace{\text{softmax}_2\left(\beta\boldsymbol{K}\boldsymbol{Q}^T\right)\boldsymbol{Q}\boldsymbol{W}_K^T}_{\textbf{Attention}} - \gamma_{\text{reg}}\underbrace{\boldsymbol{\mathcal{D}}\left(\text{softmax}\left(\frac{1}{2}\text{diag}(\boldsymbol{K}\boldsymbol{K}^T)\right)\right)\boldsymbol{K}\boldsymbol{W}_K^T}_{\textbf{Regularization}}, \qquad (12)$$

where the first and second terms are named as *attention* and *regularization* term, respectively.

Our theoretical observations offer valuable insights into implicit energy minimization by modifying the forward path of cross-attention. Inspired by these observations, we have transplanted the energy-based cross-attention layers to pre-trained text-to-image diffusion models. We first illustrate the challenges associated with adopting EBCA in a deep denoising auto-encoder and subsequently demonstrate how to overcome them in practice.

If a single recurrent Transformer block were available, a single energy function would be minimized for a given cross-attention layer by recurrently passing the updated context $\boldsymbol{C}_{n+1}$. However, in practical applications, there are multiple layer- and time-dependent energy functions in conventional deep denoising auto-encoder, which makes it infeasible to minimize each energy function individually. To address this challenge, we implicitly minimize a nested hierarchy of energy functions in a single forward pass based on our theoretical observations. More details are followed.

**Algorithms.** At a given time step $t$, we initialize the context vectors $\boldsymbol{C}_{1,t}$ with text embeddings $\boldsymbol{C}_{CLIP}$ obtained from a pre-trained CLIP. Then, within the $n$-th cross-attention layer with $\boldsymbol{C}_{n,t}$ ($1 \leq n \leq L$), we compute the updated context vectors $\boldsymbol{C}_{n+1,t}$ using the gradients in Theorem 1 and (12). We then *cascade* $\boldsymbol{C}_{n+1,t}$ to the next $(n+1)$-th cross-attention layer. We do not forward the final context $\boldsymbol{C}_{L+1,t}$ at time $t$ to time $t+1$, as the distributions of $\boldsymbol{Q}_t$ and $\boldsymbol{Q}_{t+1}$ can be significantly different due to the reverse step of the diffusion model. Instead, we reinitialize $\boldsymbol{C}_{1,t+1}$ with $\boldsymbol{C}_{CLIP}$. The pseudo-code for the proposed framework is provided in appendix B.

The sequence of energy-based cross-attention layers is designed to minimize *nested* energy functions. While a single-layer update in a single step may not lead to optimal convergence, the proposed layer-cascade context optimization can synergistically improve the quality of context vectors. Notably, our proposed method does not incur additional computational costs in practice since the gradient term in Theorem 1 relies on the existing keys, queries, and attention maps in the cross-attention layer, which can be obtained for free in the forward path.

### 3.2 Energy-based Composition of Queries

In addition to the above key advantages, the cross-attention space EBMs shed new light on the zero-shot compositional generation. Recent studies [8, 10] have demonstrated that the underlying distributions of EBMs can be combined to represent different compositions of concepts. For example, given a data point $\boldsymbol{x}$ and independent concept vectors $\boldsymbol{c}_1, \ldots, \boldsymbol{c}_n$, the posterior likelihood of $\boldsymbol{x}$ given a conjunction of specific concepts is equivalent to the product of the likelihood of each individual concept as follows:

$$p(\boldsymbol{x} \mid \boldsymbol{c}_1, \ldots, \boldsymbol{c}_n) = \prod_i p(\boldsymbol{x} \mid \boldsymbol{c}_i) \propto e^{-\sum_i \mathrm{E}(\boldsymbol{x} \mid \boldsymbol{c}_i)}, \tag{13}$$

where each $\mathrm{E}(\boldsymbol{x} \mid \boldsymbol{c}_i)$ represent independently trained EBM with concept code $\boldsymbol{c}_i$. While it is an appealing solution to the controllable generation, it is notoriously difficult to train EBMs [9, 27] in a way to make themselves scalable to high-resolution image generation. Instead of directly training EBMs in pixel space, we leverage the cross-attention space EBMs and the generative power of state-of-the-art pre-trained DMs to achieve high-fidelity compositional generations.

More specifically, assume that we have *main* context vectors $\boldsymbol{C}_1$ embedded from a main prompt, e.g. `"a castle next to a river"`, and a set of independent *editing* context vectors, $\boldsymbol{C} = \{\boldsymbol{C}_2, \ldots, \boldsymbol{C}_M\}$, each embedded from different editorial prompt, e.g. `"monet style"`, `"boat on a river"`, etc. Then, we define the keys $\boldsymbol{K}_{l,s}$ for context $s$ within a cross-attention layer of index $l$ as $\boldsymbol{K}_{l,s} = \boldsymbol{C}_s \boldsymbol{W}_{K,l}, \forall s \in \{1, 2, \ldots, M\}$. The index $l$ will be omitted for notational simplicity. Then, for a given key $\boldsymbol{K}_s$, we consider the energy function in cross-attention space w.r.t *queries* $\boldsymbol{Q}$:

$$\mathrm{E}(\boldsymbol{Q}; \boldsymbol{K}_s) = \frac{1}{2} \operatorname{diag}(\boldsymbol{Q}\boldsymbol{Q}^T) - \sum_{i=1}^{P^2} \operatorname{logsumexp}(\boldsymbol{K}_s \boldsymbol{q}_i^T, \beta), \tag{14}$$

which is directly motivated by (3). We then introduce the compositional energy function $\hat{\mathrm{E}}$, for the concept conjunction of $\boldsymbol{C}$ as in (13) and the updated rule as done in (6):

$$\hat{\mathrm{E}}(\boldsymbol{Q}; \{\boldsymbol{K}_s\}_{s=1}^M) = \frac{1}{M} \sum_{s=1}^M \mathrm{E}(\boldsymbol{Q}; \boldsymbol{K}_s)$$
$$= \frac{1}{2} \operatorname{diag}(\boldsymbol{Q}\boldsymbol{Q}^T) - \frac{1}{M} \sum_{s=1}^M \sum_{i=1}^{P^2} \operatorname{logsumexp}(\boldsymbol{K}_s \boldsymbol{q}_i^T, \beta), \tag{15}$$

$$\boldsymbol{Q}_{TF}^{new} = \frac{1}{M} \sum_{s=1}^M \alpha_s \operatorname{softmax}_2\left(\beta \boldsymbol{Q}_{TF} \boldsymbol{K}_{TF,s}^T\right) \boldsymbol{V}_{TF}, \tag{16}$$

where $\boldsymbol{Q}_{TF}$, $\boldsymbol{K}_{TF,s}$ and $\boldsymbol{V}_{TF}$ directly follows the definition in (6) and the degree and direction of $s$-th composition can be controlled for each concept individually by setting the scalar $\alpha_s$, with $\alpha_s < 0$ for concept negation [8]. Note that this is exactly a linear combination of transformer cross-attention outputs from different contexts with $\beta = 1/\sqrt{d_H}$. We refer to this process as the Energy-based Composition of Queries (EBCQ).

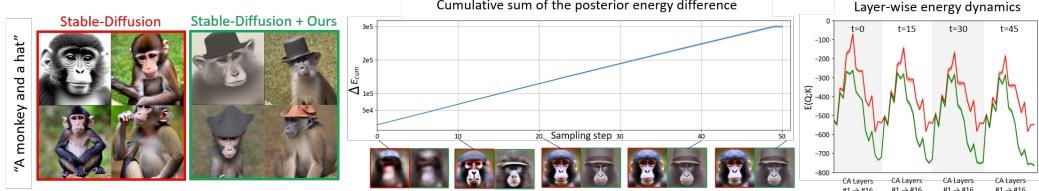

Figure 3: Energy analysis for multi-concept image generation. The right graph displays $E(Q; K)$ across 16 cross-attention layers within each sampling step. The red and green lines correspond to the Stable Diffusion and ours, respectively. Mean and standard deviation are calculated and displayed. The bottom plot shows the cumulative sum of the posterior energy difference between Stable Diffusion and the proposed method, and intermediate denoised estimates are displayed together.

This update rule implies that the compositional energy can be minimized implicitly through modifications to the forward path of EBCA, thereby guiding the integration of semantic information conveyed by editorial prompts. Notably, this is a significant departure from the existing energy-based compositional methods [8, 26, 9]. Specifically, no training or fine-tuning is required. Instead, cross-attention outputs of the main and editorial prompts are simply obtained in parallel, averaged, and propagated to the next layer. Moreover, the introduction of EBMs in cross-attention space is orthogonal to [22], which conducts compositional generation by treating a pre-trained diffusion model $\epsilon_{\theta*}$ itself as an implicitly parameterized EBM.

# 4 Experimental Results

To verify our claim of energy minimization via modified cross-attention, we conduct various experiments in text-guided image generation to verify semantic alignment, namely (1) multi-concept generation [22, 11], (2) text-guided image inpainting [23, 41], and (3) compositional generation [12, 1, 28] which includes real and synthetic image editing. In this work, while the first and third applications address similar challenges, we categorize them separately based on the implementation derived from the energy-based interpretation. Specifically, the Energy-based Bayesian Context Update (EBCU) is applied to every task, and the Energy-based Composition of Queries (EBCQ) is additionally leveraged in the third task. Note that all applications have been done without additional training.

**Setup.** The proposed framework can be widely mounted into many state-of-the-art text-to-image DMs due to its unique functionality of context updates and cross-attention map compositions. Here, we verify the effectiveness of energy-based cross-attention with Stable Diffusion (SD) [31]. The SD is an LDM that is pre-trained on a subset of the large-scale image-language pair dataset, LAION-5B [34] followed by the subsequent fine-tuning on the LAION-aesthetic dataset. For the text embedding, we use a pre-trained CLIP [29] model following the Imagen [33]. The pre-trained SD is under the creativeML OpenRAIL license. Detailed experimental settings are provided in the appendix.

## 4.1 Analysis on Energy during the sampling

We perform a comprehensive analysis on (8) and (9) during the forward path through the modified cross-attention, which offers insights into the energy behavior for real applications. Specifically, we examine the energy dynamics involved in the multi-concept image generation that is straightforward and could be readily applied to other applications with minimal effort. We record computed energy values along layers and sampling steps 30 times with a fixed text prompt and then display them in Figure 3 with the generated samples, where red and green lines denote the energy involved with the Stable-Diffusion (SD) and the proposed method, respectively. For each sampling step block that is alternately shaded, the energy values are plotted through 16 cross-attention layers.

Across all layers and sampling steps, the energy associated with the proposed method remains consistently lower than that of the SD. This is in line with the semantic alignment between intermediate denoised estimates and the given text prompt. In both cases of the SD and the proposed method, $E(\boldsymbol{Q}; \boldsymbol{K})$ decreases over sampling steps. This implies that the iterative refinement of the updated query carried over to subsequent sampling steps, resulting in a better match to the given context. Note that the proposed method even achieves lower energy with the EBCU. More analyses are provided in the appendix E including the ablation study.

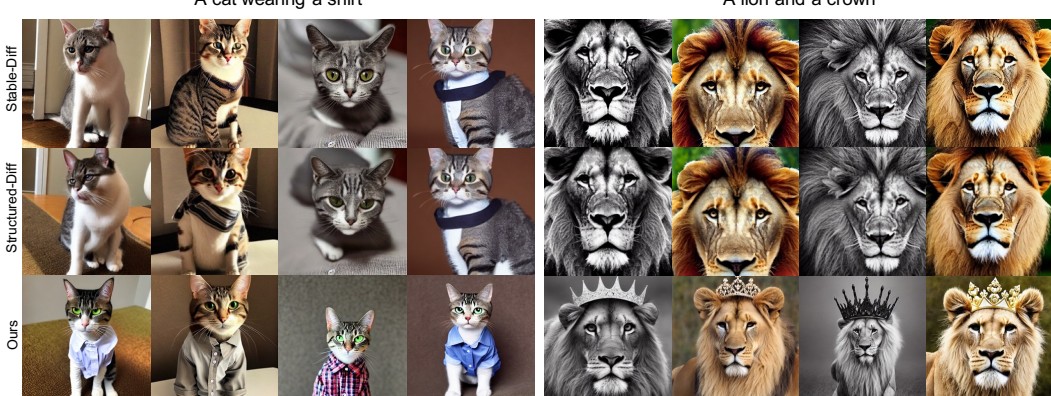

Figure 4: Multiconcept generation comparison. Each row indicates the Stable Diffusion, the Structured Diffusion, and the proposed method applied to the Stable Diffusion, respectively. Generated samples in the same column used the same text prompt and the same random seed.

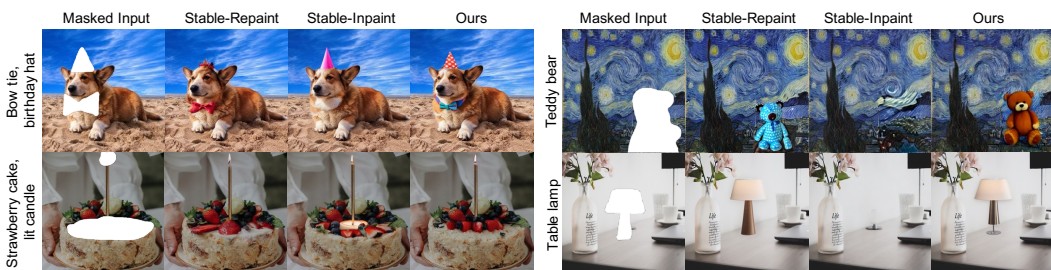

Figure 5: Text-guided inpainting comparison. The masked region is conditionally generated by given text-prompt positioned on the left-end of each row.

## 4.2 Multi-Concept Generation

We empirically demonstrate that the proposed framework alleviates the catastrophic neglect and attribute binding problems defined in the existing literature [4]. As shown in Figures 3 and 4, the EBCU effectively mitigates these problems by promoting attention to all relevant concepts in the prompt. Specifically, the regularization term introduced by the prior energy prevents a single context from dominating the attention, while the attention term updates the context vectors in the direction of input queries, improving semantic alignment and facilitating the retrieval of related information.

## 4.3 Text-guided Image Inpainting

In addition, we have evaluated the efficacy of the proposed energy-based EBCU on a text-guided inpainting task. Although existing state-of-the-art DMs such as DALLE and SD can be employed for inpainting by exploiting the ground-truth unmasked area [23], they usually require computationally expensive fine-tuning and tailored data augmentation strategies [23, 41]. In contrast, as shown in Figure 5, our proposed energy-based framework significantly enhances the quality of inpainting without any fine-tuning. Specifically, we incorporate the energy-based cross-attention into the Stable-Repaint (SR) and Stable-Inpaint (SI) models, which can struggle to inpaint multiple objects (e.g., `birthday hat` and `bow tie`) or unlikely combinations of foreground and background objects (e.g., Teddy bear on the Starry Night painting). In contrast, the proposed approach accurately fills in semantically relevant objects within the target mask region.

## 4.4 Compositional Generation

We demonstrate that the proposed framework improves the controllability and compositionality of DMs, which is a significant challenge for generative models. To assess this, we split the task into synthetic and real image editing.

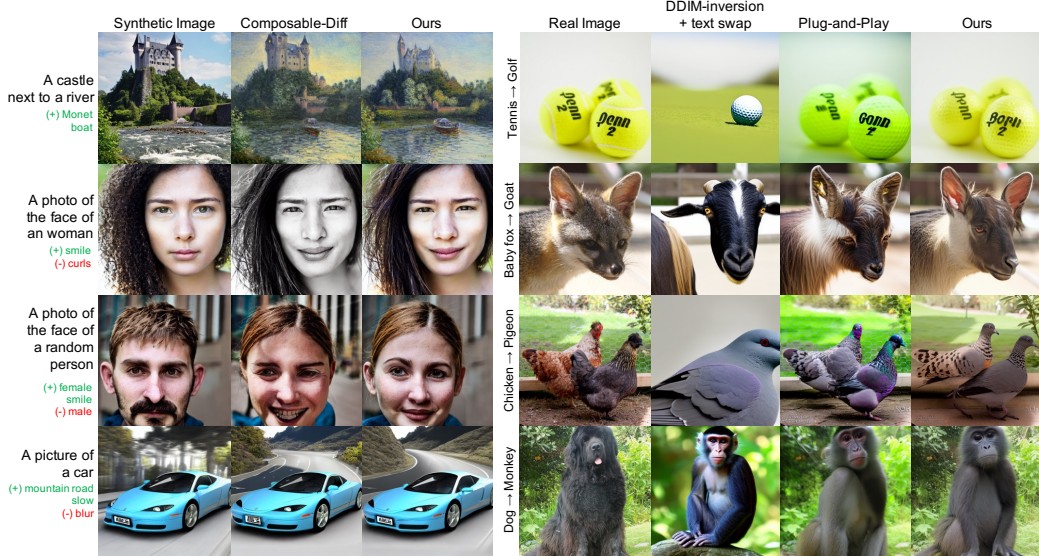

Figure 6: Image editing comparison. The left three columns show examples of synthetic image editing, while the right four columns show examples of real image editing. To ensure a fair comparison, the same editing prompt, positioned on the left-end of each row, is given to all methods being compared.

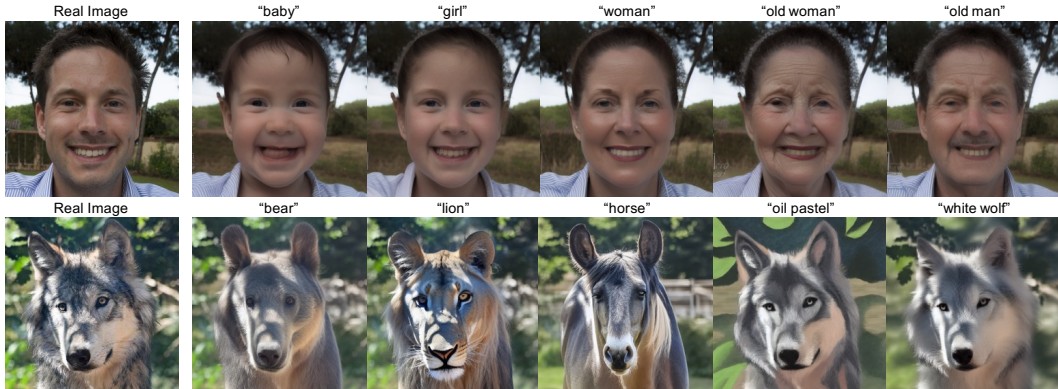

Figure 7: Real image editing with multiple editing prompts. Given a real image at the left-end of each row, the proposed method allows robust image editing with different prompts over each sample.

**Synthetic image editing.** Text-to-image DMs provide impressive results, but it is difficult to generate images perfectly aligned with user intentions [1]. Although modifying prompts can guide generation, it can also introduce unintended changes in the generated content. Our results, shown in Figure 5, demonstrate that the proposed method can edit image contents while maintaining the original-prompt-related identity of the generated images thanks to the EBCU, which continuously harmonizes the latent representations and text prompts. While composable-diff [22] can generate compositions, they often fail to preserve the original properties of images, such as the color in the second row, or fail to compose the desired concept, such as the boat in the first row.

**Real image editing.** We demonstrate that the proposed framework can also edit real images, which is especially challenging as existing methods often dramatically alter input content or introduce unexpected variations. To do this, we integrate our framework with DDIM inversion [35, 25], which inverts images with meaningful text prompts into the domain of pre-trained diffusion models. First, we use the image captioning network BLIP [20] to automatically caption the interested image, following [28]. Next, we obtain the inverted noise latents of the input image using Diffusion Pivotal Inversion [25]. Then, we apply EBCQ and EBCU while denoising the inverted latents. The optimized unconditional textual embedding vector is also targeted for additional EBCUs. The results in Figure 6 demonstrate that our method achieves better editing performance by avoiding undesired changes.

**Quantitative comparisons.** We further conducted a comparative analysis of the proposed framework against several state-of-the-art diffusion-based image editing methods [24, 12, 25, 28]. For our evaluations, we focus on two image-to-image translation tasks. The first one is an animal transition task, which translates (1) cat → dog, (2) horse → zebra, and (3) adding glasses to cat input images (cat → cat with glasses). The second one is a human transition task, which translates (1) woman → man, (2) woman → woman with glasses, and (3) woman → man with glasses. Source images are retrieved from the LAION 5B dataset [34] and CelebA-HQ [15]. Motivated by [39, 28], we measure CLIP Accuracy and DINO-ViT structure distance. Experimental details are provided in the appendix E including explanations of baselines, datasets, metrics, and hyperparameter configurations.

Table 1: (Animal transition) Comparison to state-of-the-art diffusion-based editing methods. Dist for DINO-ViT Structure distance. Baseline results are from [28].

| Method | (a) Cat → Dog | | (b) Horse → Zebra | | (c) Cat → Cat w/ glasses | |
|---|---|---|---|---|---|---|
| | CLIP Acc (↑) | Dist (↓) | CLIP Acc (↑) | Dist (↓) | CLIP Acc (↑) | Dist (↓) |
| SDEdit [24] + word swap | 71.2% | 0.081 | 92.2% | 0.105 | 34.0% | 0.082 |
| DDIM + word swap | 72.0% | 0.087 | **94.0**% | 0.123 | 37.6% | 0.085 |
| prompt-to-prompt [12] | 66.0% | 0.080 | 18.4% | 0.095 | 69.6% | 0.081 |
| pix2pix-zero [28] | 92.4% | 0.044 | 75.2% | 0.066 | 71.2% | **0.028** |
| Stable Diffusion + ours | **93.7**% | **0.040** | 90.4% | **0.061** | **81.1**% | 0.052 |

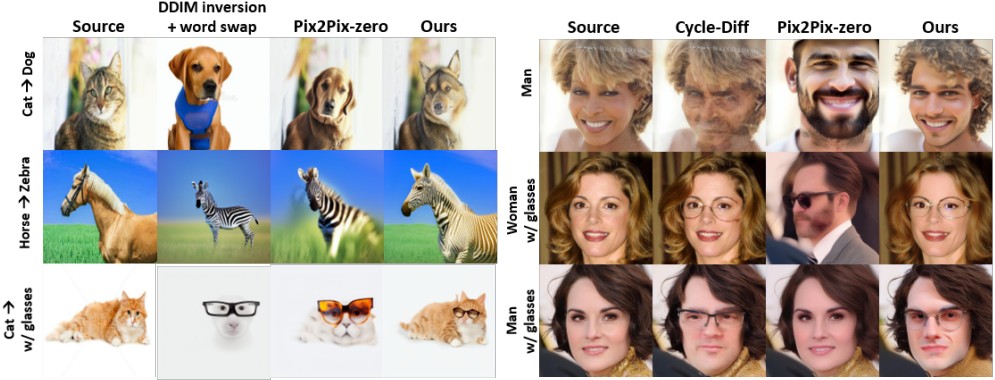

Figure 8: Real image editing on (**a**) animal transition and (**b**) human transition tasks.

Table 1 and 2 in the appendix show that the proposed energy-based framework gets a high CLIP-Acc while having low Structure Dist. It implies that the proposed framework can perform the best edit while still retaining the structure of the original input image. While DDIM + word swap records remarkably high CLIP-Acc in horse → zebra task, Figure 8 shows that such improvements are based on unintended changes in the overall structure. More visualizations are provided in the appendix E.

## 5  Conclusion, Limitations and Societal Impacts

**Conclusion.** In this work, we formulated the cross-attention with an energy perspective and proposed the EBCU, a modified cross-attention that could implicitly minimize the energy in the latent space. Furthermore, we proposed EBCQ which is inspired by energy-based formulation of multiple text composition. The proposed method is versatile as shown by multiple applications, theory-grounded, easy to implement, and computationally almost free.

**Limitations and Societal Impacts.** The framework presented in this study generates images based on user intentions, which raises concerns regarding potential misuse for creating deepfakes or other forms of disinformation. It is crucial to ensure that these methods are implemented ethically and regulated appropriately. Additionally, while the framework performs well across various tasks, it requires pre-trained deep models, rendering it challenging to apply to out-of-domain datasets, such as medical images.

**Acknowledgments**

This research was supported by the KAIST Key Research Institute (Interdisciplinary Research Group) Project, National Research Foundation of Korea under Grant NRF-2020R1A2B5B03001980, Field-oriented Technology Development Project for Customs Administration through National Research Foundation of Korea(NRF) funded by the Ministry of Science & ICT and Korea Customs Service(**NRF-2021M3I1A1097938**), Korea Medical Device Development Fund grant funded by the Korea government (the Ministry of Science and ICT, the Ministry of Trade, Industry and Energy, the Ministry of Health & Welfare, the Ministry of Food and Drug Safety) (Project Number: 1711137899, KMDF_PR_20200901_0015), Institute of Information & communications Technology Planning & Evaluation (IITP) grant funded by the Korea government(MSIT) (No.2019-0-00075, Artificial Intelligence Graduate School Program(KAIST)), Institute of Information & communications Technology Planning & Evaluation (IITP) grant funded by the Korea government(MSIT) (No.2021-0-02068, Artificial Intelligence Innovation Hub), Institute of Information & communications Technology Planning & Evaluation (IITP) grant funded by the Korea government (MSIT) (No. RS-2023-00233251, System3 reinforcement learning with high-level brain functions), and the National Research Foundation of Korea (NRF) grant funded by the Korea government (MSIT) (NRF-2019M3E5D2A01066267).

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

# A  Proof of Theorem 1

In computing a derivative of a scalar, vector, or matrix with respect to a scalar, vector, or matrix, we should be consistent with the notation. In this paper, we follow the denominator layout convention as described in [42]. To make the paper self-contained, we briefly introduce the denominator layout and associate calculus.

The main motivation for using the denominator layout is from the derivative with respect to the matrix. More specifically, for a given scalar $c$ and a matrix $\boldsymbol{W} \in \mathbb{R}^{m \times n}$, according to the denominator layout, we have

$$\frac{\partial c}{\partial \boldsymbol{W}} = \begin{bmatrix} \frac{\partial c}{\partial w_{11}} & \cdots & \frac{\partial c}{\partial w_{1n}} \\ \vdots & \ddots & \vdots \\ \frac{\partial c}{\partial w_{m1}} & \cdots & \frac{\partial c}{\partial w_{mn}} \end{bmatrix} \in \mathbb{R}^{m \times n}. \tag{17}$$

Furthermore, this notation leads to the following familiar result:

$$\frac{\partial \boldsymbol{a}^\top \boldsymbol{x}}{\partial \boldsymbol{x}} = \frac{\partial \boldsymbol{x}^\top \boldsymbol{a}}{\partial \boldsymbol{x}} = \boldsymbol{a}. \tag{18}$$

Accordingly, for a given scalar $c$ and a matrix $\boldsymbol{W} \in \mathbb{R}^{m \times n}$, we can show that

$$\frac{\partial c}{\partial \boldsymbol{W}} := \text{UnVec} \left( \frac{\partial c}{\partial \text{Vec}(\boldsymbol{W})} \right) \in \mathbb{R}^{m \times n}, \tag{19}$$

in order to be consistent with (17), where $\text{Vec}()$ and $\text{UnVec}()$ refer to the vectorization operation and its reverse, respectively. Under the denominator layout notation, for given vectors $\boldsymbol{x} \in \mathbb{R}^m$ and $\boldsymbol{y} \in \mathbb{R}^n$, the derivative of a vector with respect to a vector given by

$$\frac{\partial \boldsymbol{y}}{\partial \boldsymbol{x}} = \begin{bmatrix} \frac{\partial y_1}{\partial x_1} & \cdots & \frac{\partial y_n}{\partial x_1} \\ \vdots & \ddots & \vdots \\ \frac{\partial y_1}{\partial x_m} & \cdots & \frac{\partial y_n}{\partial x_m} \end{bmatrix} \in \mathbb{R}^{m \times n}. \tag{20}$$

Then, the chain rule can be specified as follows:

$$\frac{\partial c(\boldsymbol{g}(\boldsymbol{u}))}{\partial \boldsymbol{x}} = \frac{\partial \boldsymbol{u}}{\partial \boldsymbol{x}} \frac{\partial \boldsymbol{g}(\boldsymbol{u})}{\partial \boldsymbol{u}} \frac{\partial c(\boldsymbol{g})}{\partial \boldsymbol{g}}. \tag{21}$$

Finally, the following property of the Kronecker delta product is useful throughout the paper [42]

$$\text{Vec}(\boldsymbol{A}\boldsymbol{B}\boldsymbol{C}) = (\boldsymbol{C}^T \otimes \boldsymbol{A})\text{Vec}(\boldsymbol{B}) \tag{22}$$

$$(\boldsymbol{A} \otimes \boldsymbol{B})^T = \boldsymbol{A}^T \otimes \boldsymbol{B}^T \tag{23}$$

Using this, we first prove the following key lemmas.

**Lemma 1.** *For a given column vector $\boldsymbol{x} \in \mathbb{R}^N$, we have*

$$\frac{\partial \log\text{sumexp}(\boldsymbol{x}, \beta)}{\partial \boldsymbol{x}} = \text{softmax}(\beta \boldsymbol{x})$$

*Proof.*

$$\frac{\partial \log\text{sumexp}(\boldsymbol{x}, \beta)}{\partial \boldsymbol{x}} = \beta^{-1} \frac{\partial \log \sum_{j=1}^N \exp(\beta x_j)}{\partial \boldsymbol{x}} = \begin{bmatrix} \frac{\exp(\beta x_1)}{\sum_{j=1}^N \exp(\beta x_j)} \\ \vdots \\ \frac{\exp(\beta x_{P2})}{\sum_{j=1}^N \exp(\beta x_j)} \end{bmatrix}$$

$$= \text{softmax}(\beta \boldsymbol{x})$$

Q.E.D. □

**Lemma 2.** *Let $\boldsymbol{k}_i$ denote the $i$-th row vector of $\boldsymbol{K} \in \mathbb{R}^{N \times d}$. Then, we have*

$$\frac{\partial \boldsymbol{k}_i \boldsymbol{k}_i^T}{\partial \boldsymbol{K}} = 2 \boldsymbol{e}_N^i (\boldsymbol{e}_N^i)^T \boldsymbol{K}$$

*where $\boldsymbol{e}_N^i$ represents a $N$-dimensional column vector where only the $i$-th entry is 1, with all other entries set to zero.*

*Proof.* First, note that the expression $\boldsymbol{k}_i^T$ can be equivalently rewritten as $\boldsymbol{K}^T \boldsymbol{e}^i$. Then, we have

$$\boldsymbol{k}_i \boldsymbol{k}_i^T = (\boldsymbol{e}_N^i)^T \boldsymbol{K} \boldsymbol{K}^T \boldsymbol{e}_N^i$$

Furthermore, using (22), we have $\boldsymbol{K}^T \boldsymbol{e}_N^i = ((\boldsymbol{e}_N^i)^T \otimes \boldsymbol{I}_d)\mathrm{VEC}(\boldsymbol{K}^T)$ so that

$$\frac{\partial \boldsymbol{K}^T \boldsymbol{e}^i}{\partial \mathrm{VEC}(\boldsymbol{K}^T)} = (\boldsymbol{e}_N^i \otimes \boldsymbol{I}_d)$$

where $\boldsymbol{I}_d$ denotes the $d \times d$ identity matrix. Using the chain rule, we have

$$\frac{\partial \boldsymbol{k}_i \boldsymbol{k}_i^T}{\partial \mathrm{VEC}(\boldsymbol{K}^T)} = \frac{\partial \boldsymbol{K}^T \boldsymbol{e}^i}{\partial \mathrm{VEC}(\boldsymbol{K}^T)} \frac{\partial \boldsymbol{k}_i \boldsymbol{k}_i^T}{\partial \boldsymbol{K}^T \boldsymbol{e}^i} = 2(\boldsymbol{e}_N^i \otimes \boldsymbol{I}_d)\boldsymbol{K}^T \boldsymbol{e}_N^i$$

Thus, we have

$$\mathrm{UNVEC}\left(\frac{\partial \boldsymbol{k}_i \boldsymbol{k}_i^T}{\partial \mathrm{VEC}(\boldsymbol{K}^T)}\right) = 2\mathrm{UNVEC}\left((\boldsymbol{e}_N^i \otimes \boldsymbol{I}_d)\boldsymbol{K}^T \boldsymbol{e}_N^i\right) = 2\boldsymbol{K}^T \boldsymbol{e}_N^i(\boldsymbol{e}_N^i)^T$$

where we again use (22). Therefore, by taking the transpose, we have

$$\frac{\partial \boldsymbol{k}_i \boldsymbol{k}_i^T}{\partial \boldsymbol{K}} = 2\boldsymbol{e}_N^i(\boldsymbol{e}_N^i)^T \boldsymbol{K}.$$

Q.E.D. $\qquad\qquad\qquad\qquad\qquad\qquad\qquad\qquad\qquad\qquad\qquad\qquad\qquad\qquad\qquad\qquad\qquad\square$

**Lemma 3.**

$$\nabla_{\boldsymbol{K}} \mathrm{logsumexp}(\boldsymbol{Q}\boldsymbol{k}_i^T, \beta) = \boldsymbol{e}_N^i(\mathrm{softmax}(\beta \boldsymbol{Q}\boldsymbol{k}_i^T))^T \boldsymbol{Q} \qquad (24)$$

*Proof.* Note that

$$\boldsymbol{Q}\boldsymbol{k}_i^T = \boldsymbol{Q}\boldsymbol{K}^T \boldsymbol{e}_N^i = \boldsymbol{Q}\boldsymbol{K}^T \boldsymbol{e}_N^i = ((\boldsymbol{e}_N^i)^T \otimes \boldsymbol{Q})\mathrm{VEC}(\boldsymbol{K}^T).$$

Hence, using (18) we have

$$\frac{\partial \boldsymbol{Q}\boldsymbol{K}^T \boldsymbol{e}^i}{\partial \mathrm{VEC}(\boldsymbol{K}^T)} = \boldsymbol{e}_N^i \otimes \boldsymbol{Q}^T \qquad (25)$$

Furthermore, using Lemma 1 we have

$$\frac{\partial \mathrm{logsumexp}(\boldsymbol{Q}\boldsymbol{k}_i^T, \beta)}{\partial \mathrm{VEC}(\boldsymbol{K}^T)} = \frac{\partial \boldsymbol{Q}\boldsymbol{k}_i^T}{\partial \mathrm{VEC}(\boldsymbol{K}^T)} \frac{\partial \mathrm{logsumexp}(\boldsymbol{Q}\boldsymbol{k}_i^T, \beta)}{\partial \boldsymbol{Q}\boldsymbol{k}_i^T}$$
$$= (\boldsymbol{e}_N^i \otimes \boldsymbol{Q}^T) \mathrm{softmax}(\beta \boldsymbol{Q}\boldsymbol{k}_i^T)$$

leading to the following:

$$\frac{\partial \mathrm{logsumexp}(\boldsymbol{Q}\boldsymbol{k}_i^T, \beta)}{\partial \boldsymbol{K}^T} = \mathrm{UNVEC}\left(\frac{\partial \mathrm{logsumexp}(\boldsymbol{Q}\boldsymbol{k}_i^T, \beta)}{\partial \mathrm{VEC}(\boldsymbol{K}^T)}\right)$$
$$= \mathrm{UNVEC}\left((\boldsymbol{e}_N^i \otimes \boldsymbol{Q}^T) \mathrm{softmax}(\beta \boldsymbol{Q}\boldsymbol{k}_i^T)\right)$$
$$= \boldsymbol{Q}^T \mathrm{softmax}(\beta \boldsymbol{Q}\boldsymbol{k}_i^T)(\boldsymbol{e}_N^i)^T$$

where we again use (22). Now, by taking the transpose, we can prove (24). Q.E.D. $\qquad\square$

**Lemma 4.**

$$\frac{\partial \mathrm{diag}(\boldsymbol{K}\boldsymbol{K}^T)}{\partial \boldsymbol{K}} = 2\boldsymbol{K} \qquad (26)$$

$$\frac{\partial \log \sum_{i=1}^N \exp(\frac{1}{2}\boldsymbol{k}_i \boldsymbol{k}_i^T)}{\partial \boldsymbol{K}} = \begin{bmatrix} [\mathrm{softmax}(\boldsymbol{x})]_1 & \cdots & 0 \\ \vdots & \ddots & \vdots \\ 0 & \cdots & [\mathrm{softmax}(\boldsymbol{x})]_N \end{bmatrix} \boldsymbol{K} \qquad (27)$$

where $\mathrm{diag}(\boldsymbol{A})$ denotes the sum of the diagonal element, i.e. trace of $\boldsymbol{A}$, and $\boldsymbol{x}_i$ is a column matrix whose $i$-th element is given by $\boldsymbol{k}_i \boldsymbol{k}_i^T/2$, and $[\cdot]_i$ dentoes the $i$-th element.

*Proof.* First,

$$c = \mathrm{diag}(\boldsymbol{K}\boldsymbol{K}^T) = \sum_{i,j} K_{ij}^2$$

where $K_{ij}$ denotes the $(i, j)$-th element. Therefore, using the denominator layout in (17), it is trivial to show that

$$\frac{\partial c}{\partial \boldsymbol{K}} = 2\boldsymbol{K}.$$

Second, let us construct a column vector $\boldsymbol{x}$ whose $i$-th element is given by $x_i := \boldsymbol{k}_i \boldsymbol{k}_i^T / 2$. Then, using Lemmas 1 and 2 and the chain rule, we have

$$\frac{\partial \log \sum_{i=1}^N \exp(\frac{1}{2} \boldsymbol{k}_i \boldsymbol{k}_i^T)}{\partial \boldsymbol{K}} = \sum_i \frac{\partial x_i}{\partial \boldsymbol{K}} \frac{\partial \operatorname{logsumexp}(\boldsymbol{x}, 1)}{\partial x_i}$$

$$= \sum_i \boldsymbol{e}_N^i (\boldsymbol{e}_N^i)^T \boldsymbol{K} [\operatorname{softmax}(\boldsymbol{x})]_i$$

$$= \sum_i \boldsymbol{e}_N^i \boldsymbol{k}_i [\operatorname{softmax}(\boldsymbol{x})]_i$$

$$= \begin{bmatrix} [\operatorname{softmax}(\boldsymbol{x})]_1 & \cdots & 0 \\ \vdots & \ddots & \vdots \\ 0 & \cdots & [\operatorname{softmax}(\boldsymbol{x})]_N \end{bmatrix} \boldsymbol{K}$$

This concludes the proof. $\qquad\square$

**Theorem 1.** *For the energy functions*

$$\mathrm{E}(\boldsymbol{Q}; \boldsymbol{K}) = \frac{\alpha}{2} \operatorname{diag}(\boldsymbol{K}\boldsymbol{K}^T) - \sum_{i=1}^N \operatorname{logsumexp}(\boldsymbol{Q}\boldsymbol{k}_i^T, \beta) \tag{28}$$

*and*

$$E(\boldsymbol{K}) = \log \sum_{i=1}^N \exp(\frac{1}{2} \boldsymbol{k}_i \boldsymbol{k}_i^T), \tag{29}$$

*the gradient of the log posterior is given by:*

$$\nabla_{\boldsymbol{K}} \log p(\boldsymbol{K} \mid \boldsymbol{Q}) = \operatorname{softmax}_2 \left(\beta \boldsymbol{K}\boldsymbol{Q}^T\right) \boldsymbol{Q} - \left(\alpha \boldsymbol{I} + \mathcal{D}\left(\operatorname{softmax}\left(\frac{1}{2} \operatorname{diag}(\boldsymbol{K}\boldsymbol{K}^T)\right)\right)\right) \boldsymbol{K}, \tag{30}$$

*Then, by using the chain rule the update rule of context vectors $\boldsymbol{C}$ is derived as follows:*

$$\boldsymbol{C}_{n+1} = \boldsymbol{C}_n + \gamma \left(\operatorname{softmax}_2 \left(\beta \boldsymbol{K}\boldsymbol{Q}^T\right) \boldsymbol{Q} - \left(\alpha \boldsymbol{I} + \mathcal{D}\left(\operatorname{softmax}\left(\frac{1}{2} \operatorname{diag}(\boldsymbol{K}\boldsymbol{K}^T)\right)\right)\right) \boldsymbol{K}\right) \boldsymbol{W}_K^T, \tag{31}$$

*where $\gamma > 0$ is a step size, and $\mathcal{D}(\cdot)$ is a vector-to-diagonal-matrix operator.*

*Proof.* Based on the Bayes' theorem, the gradient of the log posterior is derived as:

$$\nabla_{\boldsymbol{K}} \log p(\boldsymbol{K} \mid \boldsymbol{Q}) = -\left(\nabla_{\boldsymbol{K}} \mathrm{E}(\boldsymbol{Q}; \boldsymbol{K}) + \nabla_{\boldsymbol{K}} \mathrm{E}(\boldsymbol{K})\right). \tag{32}$$

Using Lemmas 1,2,3 and 4, we have

$$\nabla_{\boldsymbol{K}} \mathrm{E}(\boldsymbol{Q}; \boldsymbol{K}) = \alpha \boldsymbol{K} - \sum_i \boldsymbol{e}_N^i (\operatorname{softmax}(\beta \boldsymbol{Q}\boldsymbol{k}_i^T))^T \boldsymbol{Q}$$

$$= \alpha \boldsymbol{K} - \operatorname{softmax}_2(\beta \boldsymbol{Q}\boldsymbol{K}^T) \boldsymbol{Q} \tag{33}$$

Second, by noting that $\mathrm{E}(\boldsymbol{K}) = \log \sum_{i=1}^N \exp(\frac{1}{2} \boldsymbol{k}_i \boldsymbol{k}_i^T)$, Lemma 4 informs us

$$\nabla_{\boldsymbol{K}} \mathrm{E}(\boldsymbol{K}) = \begin{bmatrix} [\operatorname{softmax}(\boldsymbol{x})]_1 & \cdots & 0 \\ \vdots & \ddots & \vdots \\ 0 & \cdots & [\operatorname{softmax}(\boldsymbol{x})]_N \end{bmatrix} \boldsymbol{K}$$

$$= \mathcal{D}\left(\operatorname{softmax}\left(\frac{1}{2} \operatorname{diag}(\boldsymbol{K}\boldsymbol{K}^T)\right)\right) \boldsymbol{K}$$

where $\mathcal{D}(\cdot)$ is a vector-to-diagonal-matrix operator that takes $N$-dimensional softmax vector as an input and returns a $N \times N$ diagonal matrix with softmax values as main diagonal entries. Therefore, one can finally obtain:

$$\nabla_{\boldsymbol{K}} \log p(\boldsymbol{K} \mid \boldsymbol{Q}) = \operatorname{softmax}_2 \left(\beta \boldsymbol{K}\boldsymbol{Q}^T\right) \boldsymbol{Q} - \left(\alpha \boldsymbol{I} + \mathcal{D}\left(\operatorname{softmax}\left(\frac{1}{2} \operatorname{diag}(\boldsymbol{K}\boldsymbol{K}^T)\right)\right)\right) \boldsymbol{K}. \tag{34}$$

By using the chain rule with $\boldsymbol{K} = \boldsymbol{C}\boldsymbol{W}_K$, the update rule of context vectors $\boldsymbol{C}$ is derived as in (11). $\qquad\square$

# B   Pseudo-code for EBCU and EBCQ

This section provides the description of the pseudocode for the proposed Energy-based Bayesian Context Update (EBCU) and Energy-based Composition of Queries (EBCQ). Algorithm 1 outlines the cascaded context propagation across cross-attention layers within the UNet model during the sampling step $t$. Note that the context is reinitialized at the beginning of each sampling step. On the other hand, Algorithm 2 details the EBCU implemented in each cross-attention layer. Specifically, the proposed EBCU provides a significant computational efficiency by reusing the similarity $\boldsymbol{Q}\boldsymbol{K}^T$, which requires computational cost $\mathcal{O}(N^2)$, to compute $\nabla_{\boldsymbol{K}} E(\boldsymbol{Q}; \boldsymbol{K})$. Consequently, there is only a small amount of additional computational overhead associated with the proposed EBCU.

---

**Algorithm 1** Context cascade at sampling step $t$

---

**Require:** $\boldsymbol{Q}_t, \boldsymbol{C}_{clip}$, UNet
1: $\boldsymbol{C}_t \leftarrow \boldsymbol{C}_{clip}$ // Re-initialize
2: **for** layer in UNet **do**
3:      **if** layer is CrossAttention **then**
4:          $\boldsymbol{Q}_t, \boldsymbol{C}_t \leftarrow \text{layer}(\boldsymbol{Q}_t, \boldsymbol{C}_t)$ // Algorithm 2
5:      **else**
6:          $\boldsymbol{Q}_t \leftarrow \text{layer}(\boldsymbol{Q}_t)$
7:      **end if**
8: **end for**
9: $\boldsymbol{Q}_{t+1} \leftarrow \boldsymbol{Q}_t$
10: **return** $\boldsymbol{Q}_{t+1}$

---

**Algorithm 2** Energy-based Bayesian Context Update (EBCU)

---

**Require:** $\boldsymbol{Q}, \boldsymbol{C}, \boldsymbol{W}_q, \boldsymbol{W}_k, \boldsymbol{W}_v, \alpha, \beta, \gamma_{\text{attn}}, \gamma_{\text{reg}}$
1: $\boldsymbol{Q}, \boldsymbol{K}, \boldsymbol{V} \leftarrow \boldsymbol{Q}\boldsymbol{W}_q, \boldsymbol{C}\boldsymbol{W}_k, \boldsymbol{C}\boldsymbol{W}_v$
2: $\boldsymbol{S} = \boldsymbol{Q}\boldsymbol{K}^T$
3: $\boldsymbol{Q} \leftarrow \text{softmax}_2(\beta\boldsymbol{S})\boldsymbol{V}$
4: $\nabla_{\boldsymbol{K}} E(\boldsymbol{Q}; \boldsymbol{K}) = \text{softmax}_2(\beta\boldsymbol{S}^T)\boldsymbol{Q}$
5: $\nabla_{\boldsymbol{K}} E(\boldsymbol{K}) = -(\alpha\boldsymbol{I} + \boldsymbol{\mathcal{D}}(\text{softmax}(\frac{1}{2}\text{diag}(\boldsymbol{K}\boldsymbol{K}^T))))\boldsymbol{K}$
6: $\Delta\boldsymbol{C} = (\gamma_{\text{attn}}\nabla_{\boldsymbol{K}} E(\boldsymbol{Q}; \boldsymbol{K}) + \gamma_{\text{reg}}\nabla_{\boldsymbol{K}} E(\boldsymbol{K}))\boldsymbol{W}_k^T$
7: $\boldsymbol{C} \leftarrow \boldsymbol{C} + \Delta\boldsymbol{C}$
8: **return** $\boldsymbol{Q}, \boldsymbol{C}$

---

Algorithm 3 outlines the pseudocode for the EBCQ implemented for $M$ given contexts. For the simplicity, we exclude the EBCU from the algorithm. Nontheless, the EBCU and the EBCQ could be leveraged together.

---

**Algorithm 3** Energy-based Composition of Queries (EBCQ)

---

**Require:** $\boldsymbol{Q}, \boldsymbol{C} = \{\boldsymbol{C}_1, ..., \boldsymbol{C}_M\}, \boldsymbol{W}_q, \boldsymbol{W}_k, \boldsymbol{W}_v, \alpha_s, \beta$
1: $\boldsymbol{Q} \leftarrow \boldsymbol{Q}\boldsymbol{W}_q$
2: **for** $s$ in $[1, ..., M]$ **do**
3:      $\boldsymbol{K}_s, \boldsymbol{V}_s \leftarrow \boldsymbol{C}_s\boldsymbol{W}_k, \boldsymbol{C}_s\boldsymbol{W}_v$
4:      $\boldsymbol{S}_s = \boldsymbol{Q}\boldsymbol{K}_s^T$
5: **end for**
6: $\boldsymbol{Q} \leftarrow \frac{1}{M}\sum_{s=1}^M \alpha_s\text{softmax}_2(\beta\boldsymbol{S}_s)\boldsymbol{V}_s$
7: **return** $\boldsymbol{Q}$

---

# C   Experimental setups

In this section, we describe detailed experimental setups for three applications including baseline method, hyper-parameter of the proposed method, and dataset if it is the case. Code: `https://github.com/EnergyAttention/Energy-Based-CrossAttention`.

## C.1   Common experimental setup

We mainly leverage pre-trained Stable Diffusion v1-5 (except Table 1: v1-4) which is provided by *diffusers*, a Python library that offers various Stable Diffusion pipelines with pre-trained models. All images are sampled for

50 steps via PNDM sampler [21] using NVIDIA RTX 2080Ti. In every experiment, we set the parameter $\alpha$ in Equation (9) to zero, focusing solely on controlling the values of $\gamma_{attn}$ and $\gamma_{reg}$. EBCU is applied to every task, and EBCQ is additionally employed in C.4.

**Different learning rate for each token** It is worth noting that the $\gamma_{attn}$ and $\gamma_{reg}$ could be expressed as vectors. In other words, if the context $C \in \mathrm{R}^{N \times d_c}$ is given, $\gamma_{attn}$ and $\gamma_{reg}$ are $N$-dimensional vectors. Hence, we have the flexibility to adjust the learning rate $\gamma_{\{\cdot\}}$, allowing us to increase or decrease the impact of certain tokens based on the user's intent. Unless otherwise noted, $\gamma_{attn}$ and $\gamma_{reg}$ is set to a constant for each text token.

**Learning rate scheduling** Since the proposed EBCU is leveraged for the diffusion model, one can readily introduce scheduling strategies for $\gamma_{attn}$ and $\gamma_{reg}$ along the sampling step $t$. We implement multiple variants such as 'constant', 'step', and 'exponential decay' as follows.

$$
\begin{array}{ll}
\text{[constant]} & \gamma(t) = \gamma_0 \\
\text{[step]} & \gamma(t) = \gamma_0 \cdot \text{ReLu}(t - \tau) \\
\text{[exp-decay]} & \gamma(t) = \gamma_0 \cdot \lambda^t
\end{array}
\tag{35}
$$

where $\gamma_0$ is the initial value, $\text{ReLu}(x) = 0$ if $x \le 0$, otherwise 1, $\tau$ denotes the temporal threshold, and $\lambda$ denotes the decay ratio. Unless stated otherwise, the scheduling strategy is set to the 'constant'.

## C.2   Multi-concept image generation

We compared the performance of the proposed method with Structured Diffusion [11] which does not require additional training as our method. We leveraged the open-sourced official implementation [1].

For the proposed method, we set the $\gamma_{attn}$ and $\gamma_{reg}$ differently for each sample within [1e-2, 1.5e-2, 2e-2]. As shown in the following ablation studies E, large $\gamma_{attn}$ tends to generate saturated images while large $\gamma_{reg}$ results in mixed/vanished contents.

We found that using different learning rates for each context token is useful for multi-concept generation, especially when a single concept tends to dominate with a constant learning rate. For example, given the main prompt `"A cat wearing a shirt"`, we set the $\gamma_{attn}$ for the `"shirt"` to 3e-2, while $\gamma_{attn}$ is set to 1.5e-2 for other tokens. We have observed that doubling the $\gamma_{attn}$ for a text token to be emphasized is sufficient to achieve balanced multi-concept image generation for most cases.

## C.3   Text-guided image inpainting

Additionally, we conducted a performance comparison between our proposed method and two alternative approaches: (**a**) Stable Inpaint[2], which fine-tunes the weights of Stable Diffusion through inpainting training, and (**b**) Stable Repaint[3], which leverages the work of Lugmayr et al. [23] on the latent space of Stable Diffusion for the inpainting task. In the case of Stable Repaint, the mask is downsized and transferred into the latent space. We applied the Energy-based Bayesian Context Update (EBCU) technique to both methods, resulting in improved results compared to their respective baselines.

**Masked EBCU.** To further enhance the performance for the inpainting task, we introduce the concept of masked Energy-based Bayesian Context Update (masked EBCU). Specifically, let $M \in \mathbb{R}^{P_l^2 \times P_l^2}$ represent a diagonal matrix where the main diagonal values are derived from the downsampled inpainting mask for the $l$-th cross-attention layer, with an output spatial size of $P_l^2$. In Equation (36), we modify the attention term (12) by incorporating the downsampled mask, effectively covering the query matrix as follows:

$$
\boldsymbol{C}_{n+1} = \boldsymbol{C}_n + \gamma \bigg( \text{softmax}_2 \left( \beta \boldsymbol{K} \boldsymbol{Q}^T \right) \boldsymbol{M} \boldsymbol{Q} - \left( \alpha \boldsymbol{I} + \boldsymbol{\mathcal{D}} \left( \text{softmax} \left( \frac{1}{2} \text{diag}(\boldsymbol{K} \boldsymbol{K}^T) \right) \right) \right) \boldsymbol{K} \bigg) \boldsymbol{W}_K^T. \tag{36}
$$

As evident in Equation (12), the attention term updates the context vectors, aligning $\boldsymbol{k}_i$ towards $\boldsymbol{q}_j, j = 1, \ldots, P_l^2$, while considering the alignment strength between each $\boldsymbol{q}_j$ and $\boldsymbol{k}_i$. However, in the inpainting task, we have prior knowledge that the context vectors should be most aligned with the semantically relevant masked regions. Therefore, we mask out unrelated background spatial representations, allowing for the context vectors to be updated with a specific focus on the masked regions. This approach facilitates the incorporation of semantic information encoded by $\boldsymbol{k}_i$ specifically into the spatial mask regions.

In our proposed method, we set different values for $\gamma_{attn}$ and $\gamma_{reg}$ for each sample, selected from the set [1e-2, 1.5e-2, 2e-2, 2.5e-2], to account for variations in the input samples.

---

[1]`https://github.com/weixi-feng/Structured-Diffusion-Guidance`
[2]`https://huggingface.co/runwayml/stable-diffusion-inpainting`
[3]`https://github.com/huggingface/diffusers/tree/main/examples/community#`
`stable-diffusion-repaint`

## C.4 Image editing via compositional generation

We present empirical evidence demonstrating the effectiveness of our energy-based framework for compositional synthetic and real-image editing. The Energy-based Bayesian Context Update (EBCU) technique can be readily applied to both the main context vector ($C_1$ in Section 3.2, $s = 1$) and editorial context vectors ($C_{s>1}$). Each EBCU operation influences the attention maps used in the Energy-based Composition of Queries (EBCQ), enhancing the conveyance of semantic information associated with each context. Note that $\alpha_s$ in (16) represents the degree of influence of the $s$-th concept in the composition. In practice, we fix $\alpha_1 = 1$ for the main context, while $\alpha_{s>1}$ is tuned within the range of (0.5, 1.0).

Let $\gamma_{attn,s}$ and $\gamma_{reg,s}$ denote the step sizes for EBCU of the $s$-th context vector. If the editing process involves changing the identity of the original image (e.g., transforming a "cat" into a "dog"), we set both $\gamma_{attn,1}$ and $\gamma_{reg,1}$ to zero. Otherwise, if the editing maintains the original identity, we choose values for $\gamma_{attn,1}$ and $\gamma_{reg,1}$ from the range of (5e-4, 1e-3), similar to $\gamma_{attn,(s>1)}$ and $\gamma_{reg,(s>1)}$. All hyperparameters, including $\alpha_s$ and $\gamma_s$, are fixed during the quantitative evaluation process (more details in Section D and Table 3).

To ensure consistent results, we maintained a fixed random seed for both real and synthetic image editing. For real image editing, we employed null-text pivotal inversion [25] to obtain the initial noise vector.

During the reverse diffusion process in Sections C.2 and C.3, we kept $\gamma$ fixed as a constant value. However, for compositional generation, we utilized step scheduling (Equation 35) for $\gamma_s$ and $\alpha_s$. After converting the initial noise vector for real images or using a fixed random seed for synthetic images, EBCU and EBCQ are applied after a threshold time $\tau_s > 0$ for the $s$-th editorial context. This scheduling strategy helps to preserve the overall structure of generated images during the editing process. In our observations, a value of $\tau_s \in [10, 25]$ generally produces satisfactory results, considering a total number of reverse steps set to 50. However, one can increase or decrease $\tau_s$ for more aggressive or conservative editing, respectively.

The exemplary real images presented in Figures 5 and 6 of the main paper were sampled from datasets such as FFHQ [17], AFHQ [5], and ImageNet [7]. For a detailed quantitative analysis, please refer to Section D.

# D   Quantitative Comparison

In this section, we conducted a comparative analysis of the proposed framework against several state-of-the-art diffusion-based image editing methods [24, 12, 25, 28], following the experimental setup of [28]. To ensure a fair comparison, all methods utilize the pre-trained Stable Diffusion v1-4, employ the PNDM sampler with an equal number of sampling steps, and adopt the same classifier-free guidance scale.

## D.1   Baseline Methods

In addition to the Plug-and-Play method discussed in the main paper, we include the following baselines for comprehensive quantitative comparison:

**SDEdit [24] + word swap.** This method introduces the Gaussian noise of an intermediate timestep and progressively denoises images using a new textual prompt, where the source word (e.g., Cat) is replaced with the target word (e.g., Dog).

**Prompt-to-prompt (P2P) [12].** P2P edits generated images by leveraging explicit attention maps from a source image. The source attention maps $M_t$ are used to inject, re-weight, or override the target maps based on the desired editing operation. These original maps act as hard constraints for the edited images.

**DDIM + word swap [25].** This method applies null-text inversion to real input images, achieving high-fidelity reconstruction. DDIM sampling is then performed using inverted noise vectors and an edited prompt generated by swapping the source word with the target.

**pix2pix-zero [28].** pix2pix-zero first derives a text embedding direction vector $\triangle c_{\text{edit}}$ from the source to the target by using a large bank of diverse sentences generated from a state-of-the-art sentence generator, such as GPT-3 [2]. Inverted noise vectors are denoised with the edited text embedding, $c + \triangle c_{\text{edit}}$, and cross-attention guidance to preserve consensus.

**Cycle-diffusion [40].** Cycle-diffusion reformulates diffusion models as deterministic mappings from a Gaussian latent code to images. It presents a DPM-Encoder that allows for encoding images into this latent space.

## D.2   Dataset

For the animal transition task, we focus on three image-to-image translation tasks: (1) translating cats to dogs (cat $\rightarrow$ dog), (2) translating horses to zebras (horse $\rightarrow$ zebra), and (3) adding glasses to cat input images (cat $\rightarrow$ cat with glasses). Following the data collection protocol of [28], we retrieve 250 relevant cat images and 213

horse images from the LAION 5B dataset [34] using CLIP embeddings of the source text description. We select images with a high CLIP similarity to the source word for each task.

For the human transition task, we focus on three image-to-image translation tasks: (1) woman → man, (2) woman → woman w/ glasses, and (3) woman → man w/ glasses. For this, we select source images of women without wearing glasses from CelebA-HQ. To ensure fair comparisons, all methods leverage the same version of Stable Diffusion, sampler, sampling steps, etc.

## D.3   Metrics

Motivated by [39, 28], we measure CLIP Accuracy and DINO-ViT structure distance. Specifically, (**a**) CLIP Acc represents whether the targeted semantic contents are well reflected in the generated images. It calculates the percentage of instances where the edited image has a higher similarity to the target text, as measured by CLIP, than to the original source text [28]. On the other hand, (**b**) structure distance [39, 38] measures whether the overall structure of the input image is well preserved. It is defined as the difference in self-similarity of the keys extracted from the attention module at the deepest DINO-ViT [3] layer.

## D.4   Details

The main context vector $C_{main}$ is encoded given a main prompt automatically generated by BLIP [20]. In addition, the editorial context vectors $C_{src}$ and $C_{tgt}$ are encoded given the text descriptions of the source and target concept, i.e. source and target prompt. For example, for a cat → dog task (cat → cat w/ glasses), the source prompt is `"cat"` (`"cat wearing glasses"`), and the target prompt is `"dog"` (`"without glasses"`). Then we apply EBCU and EBCQ based on the obtained context vectors. Please refer to Table 3 for the hyperparameter configurations.

## D.5   Results

Table 1 and 2 show that the proposed energy-based framework gets a high CLIP-Acc while having low Structure Dist. It implies that the proposed framework can perform the best edit while still retaining the structure of the original input image. This is a remarkable result considering that the proposed framework is not specially designed for the real-image editing task. Moreover, the proposed framework does not rely on the large bank of prompts and editing vector $\triangle c_{edit}$ [28] which can be easily incorporated into our method.

While DDIM + word swap records remarkably high CLIP-Acc in horse → zebra task, Figure 9 and 16 show that such improvements are based on unintended changes in the overall structure. Table 3 summarizes the hyperparameter settings for each task. Examples of results are presented in Figure 17 and 16.

Table 2: (Human transition) Comparison to state-of-the-art diffusion-based editing methods. Dist for DINO-ViT Structure distance.

| Method | (a) Woman → Man | | (b) Woman → Woman w/ glasses | | (c) Woman → Man w/ glasses | |
|---|---|---|---|---|---|---|
| | CLIP Acc (↑) | Dist (↓) | CLIP Acc (↑) | Dist (↓) | CLIP Acc (↑) | Dist (↓) |
| CycleDiffusion [40] | 57.0% | 0.156 | 74.0% | **0.018** | 63.5% | 0.237 |
| Pix2pix-zero | 93.0% | 0.043 | 66.0% | 0.024 | 93.9% | 0.052 |
| Stable Diffusion + ours | **94.0**% | 0.035 | 99.0% | 0.029 | **96.0**% | **0.037** |

Table 3: Hyperparameter configurations for each editing task. Each task index comes from Table 1. $\gamma_{attn,main} = 0$ and $\gamma_{reg,main} = 0$ as mentioned in section C.4. Note that $\alpha_{src} < 0$ for the concept negation (related ablation study in Figure 11). $\tau_s$ denotes the warm-up period for step scheduling in (35) and Section C.4.

| Task | $\alpha_{src}$ | $\alpha_{tgt}$ | $\gamma_{\cdot,main}$ | $\gamma_{attn,src}$ | $\gamma_{reg,src}$ | $\gamma_{attn,tgt}$ | $\gamma_{reg,tgt}$ | $\tau_s$ |
|---|---|---|---|---|---|---|---|---|
| (**a**) | -0.65 | 0.75 | 0 | 5e-4 | 5e-4 | 6e-4 | 6e-4 | 25 |
| (**b**) | -0.5 | 0.6 | 0 | 4e-4 | 4e-4 | 5e-4 | 5e-4 | 15 |
| (**c**) | -0.6 | 0.7 | 0 | 1e-3 | 1e-3 | 1e-3 | 1e-3 | 17 |

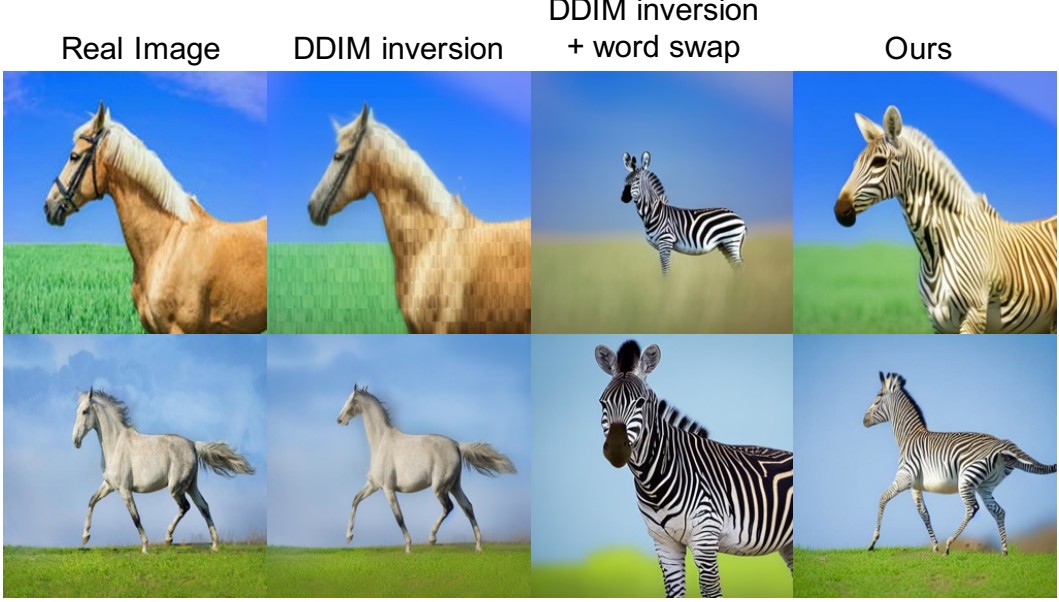

Figure 9: Image editing comparison with DDIM-inversion. Generated samples by DDIM-inversion with word swap readily deviate the original data contents, while the proposed method avoids undesired changes.

# E Ablation study and more results

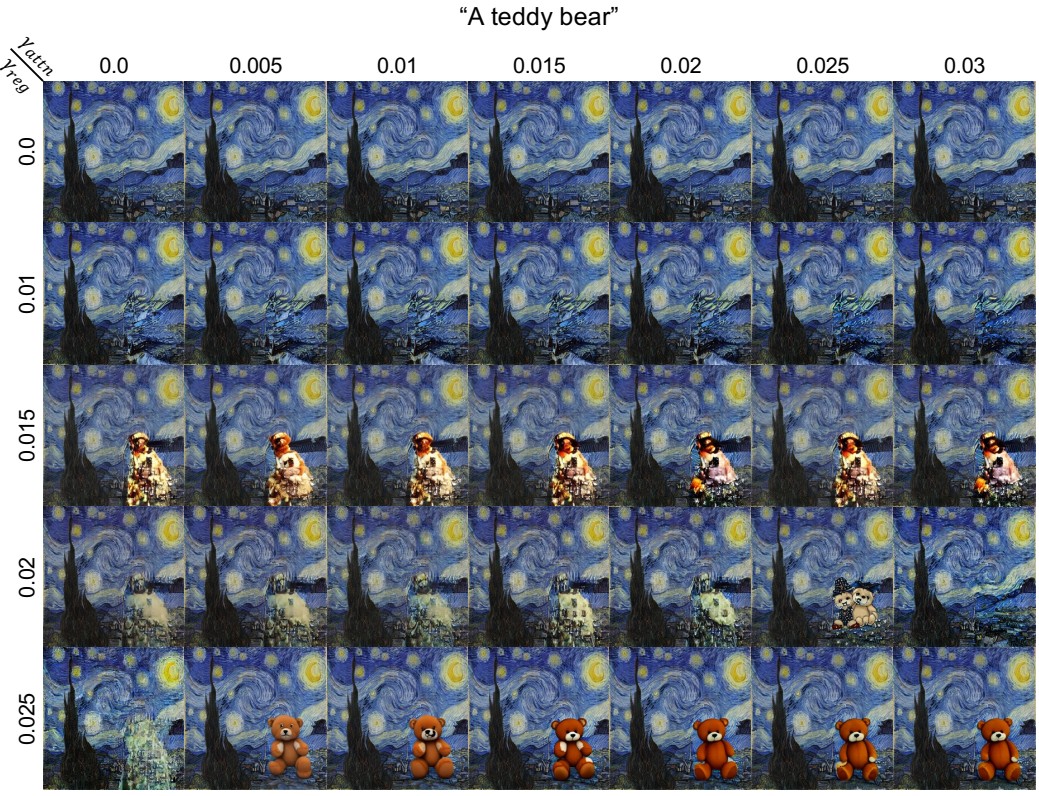

Figure 10: Ablation results for $\gamma_{attn}$ and $\gamma_{reg}$. All samples are generated from the same random noise.

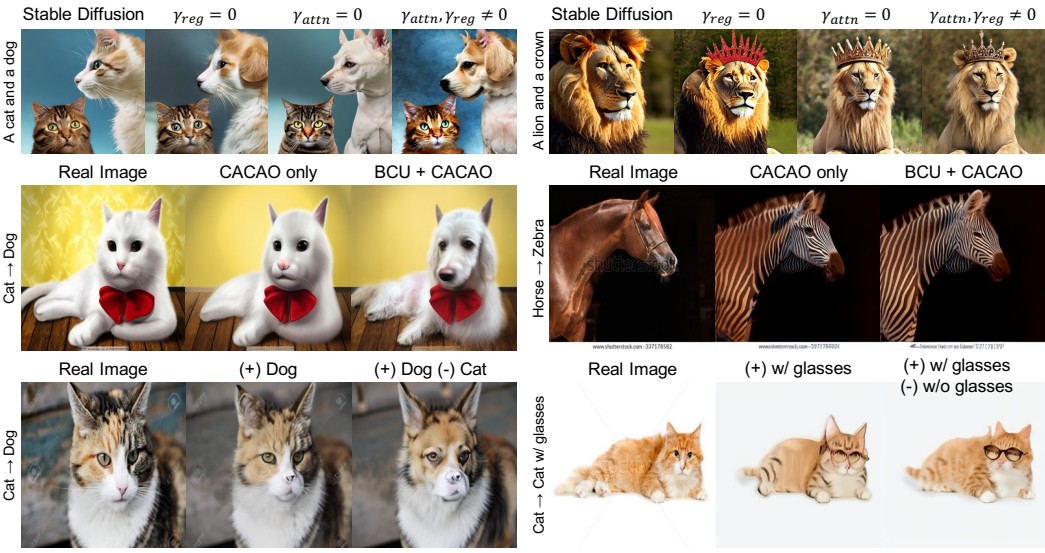

Figure 11: Ablation study results. The first row shows multi-concept generation examples with varying $\gamma_{attn}$ and $\gamma_{reg}$, while the second row shows real image editing examples with varying the usage of EBCU and EBCQ. The last row shows the effect of negative prompt for the image editing application.

**Attention and regularization terms.** To access the degree of performance improvement attained by the proposed EBCU, we conducted an ablation study for the attention and the regularization terms by regulating $\gamma_{attn}$ and $\gamma_{reg}$ for the text-guided image inpainting (Figure 10) and the multi-concept image generation (Figure 11). From the Figure 10, we can observe that the desired content is generated when proper range of $\gamma_{attn}$ and $\gamma_{reg}$ are given. Specifically, once $\gamma_{reg}$ is set to a valid value, the EBCU consistently generate a "teddy bear" with various $\gamma_{attn}$, otherwise it generates background or imperfect objects. This result emphasizes the role of the introduced prior energy $\mathrm{E}(\boldsymbol{K})$. Furthermore, the $\gamma_{attn}$ also affects to the context alignment of the generated sample (for instance $\gamma_{attn} = 0.025$ and $\gamma_{reg} = 0.02$), which highlights the importance of the introduced conditional energy function $\mathrm{E}(\boldsymbol{Q}; \boldsymbol{K})$. The same evidences could be found in the first row in Figure 11 which are the multi-concept image generation examples.

**Synergy between EBCU and EBCQ.** While both EBCU and EBCQ are designed from the common energy-based perspective, each operation is originated from different energy functions $\mathrm{E}(\boldsymbol{K}; \boldsymbol{Q})$ and $\hat{\mathrm{E}}(\boldsymbol{Q}; \{\boldsymbol{K}_s\}_{s=1}^M)$, respectively. This fact suggests the synergistic energy minimization by combining the EBCU and EBCQ, which could further improve the text-conditional image generation. To investigate this further, we conducted an ablation study using a real image editing application. Specifically, we compared the editing performance when solely utilizing EBCU and when combining EBCU with EBCQ. The second row in Figure 11 is the result of the ablation study that shows fully-compatibility of the EBCU and EBCQ. Importantly, the incorporation of the EBCU improves the quality of the generated images. While the EBCQ alone effectively captures the context of the given editing concept, the addition of EBCU enhances the fine-grained details in the generated outputs.

**Importance of concept negation.** Remark that a negative $\alpha_s$ in (16) denotes the negation of given editing prompt. We empirically observed that the concept negation may significantly contribute to the performance of compositional generation. Specifically, for the image-to-image translation task in Table 1, we apply both positive and negative guidance with the target (e.g. Dog) *and* source (e.g. Cat) concepts, respectively, following the degree of guidance denoted in Table 3. The third row in Figure 11 shows the impacts of source concept negation in the image-to-image translation task. While the positive guidance alone may fail to remove the source-concept-related features, e.g. eyes of the Cat, the negative guidance removes such conflicting existing attributes. This implies that the proposed framework enables useful arithmetic of multiple concepts for both real and synthetic image editing.

**Prior energy and $\alpha$.** While $\frac{\alpha}{2}\operatorname{diag}(\boldsymbol{K}\boldsymbol{K}^T)$ in (8) penalizes norm of each context vectors uniformly, the proposed prior energy function $\hat{\mathrm{E}}(\boldsymbol{K})$ adaptively regularizes the smooth maximum of $\|\boldsymbol{k}_i\|$. Intuitively, adaptive penalization prevents the excessive suppression of context vectors, potentially resulting in images that are more semantically aligned with a given context. To demonstrate the effectiveness of adaptive penalization in the prior energy function, we conducted a multi-concept image generation task with varying $\alpha$ in (11) from 0 to 1, while fixing other hyperparameters. Figure 12 illustrates the gradual disappearance of salient contextual elements in the generated images depending on the change of $\alpha$. Specifically, the crown is the first to diminish, followed by

subsequent context elements, with the lion being the last to vanish with $\alpha = 1$. This result highlights the validity of the adaptive penalization for the context vectors which stems from the prior energy function.

**Context shift analysis.** Since the proposed framework updates the initial context vector, one may be concerned that the optimized vector may be severely shifted from the original context so that it loses the intended semantic meaning. Here, we emphasize that the change of context vector is adaptive, not an invalid shift, because the proposed BCU allows an adaptive propagation of the context vector through UNet, resulting in a context vector that is better aligned with $Q_t$ on cross-attention space.

To further support our claim, we use the updated $C_T$ ($T$ for a number of sampling time steps) from the proposed method as a fixed context vector instead of $C_{\text{clip}}$ and perform multi-concept generation using the conventional cross-attention operation. Since the energy functions are defined differently for each sampling time step, using $C_T$ as a fixed context for each sampling time step may result in low-quality samples. However, this approach allows us to evaluate whether the updated context contains the correct semantics of the given textual conditions, in contrast to $C_{\text{clip}}$. Figure 13a demonstrates that the updated context vector does indeed capture the correct semantics of the given textual conditions (e.g., both black horse and yellow room).

**Multi-step update.** In order to better understand the context update, we implement a multistep context update for comparison while keeping the propagation of $C$ to subsequent layers. For the multi-concept generation and inpainting tasks, we observe that this multiple update of the context vector actually improves the quality of generated images (Figure 13b). This is in the same line as our perspective that one forward path of context vector is equivalent to one-step gradient descent for energy function. That being said, we also observe that the multiple context update is relatively computationally expensive and a single-step update is usually sufficient for improved performance. Therefore, we decide to use a single-step update with context propagation to subsequent layers. We believe these results may lay down foundation for further research on understanding the alignment between latent image representations and context embeddings.

$\alpha$=0.0 $\qquad\qquad\qquad\qquad\qquad$ $\alpha$=0.5 $\qquad\qquad\qquad\qquad\qquad$ $\alpha$=1.0

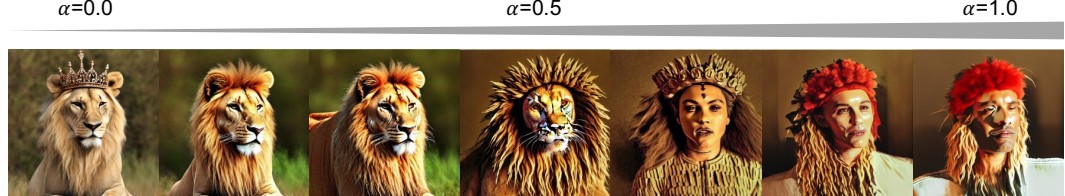

Figure 12: Generated samples with varying $\alpha$ values. As $\alpha$ increases, the generated images progressively deviate from the intended context, `"A lion and a crown"`.

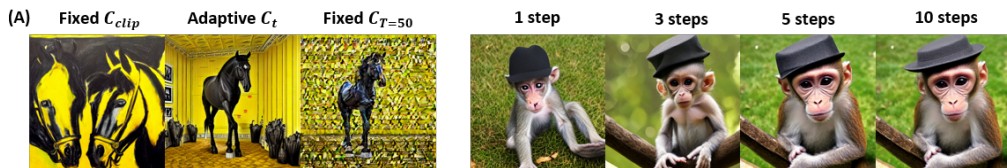

Figure 13: (**a**) Muti-concept generation results with fixed and adaptive context embeddings. For the last column, updated context via the BCU at the final timestep is given as a fixed context embedding. (**b**) Multi-concept image generation results in a different number of context updates. $\gamma_{attn}$ and $\gamma_{reg}$ are reduced in proportion to the number of updates.

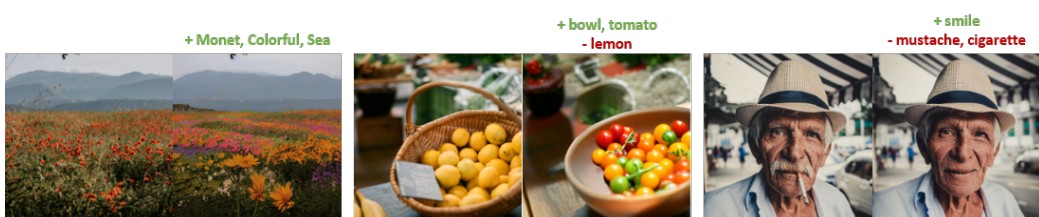

Figure 14: Further results for compositional real-image editing.

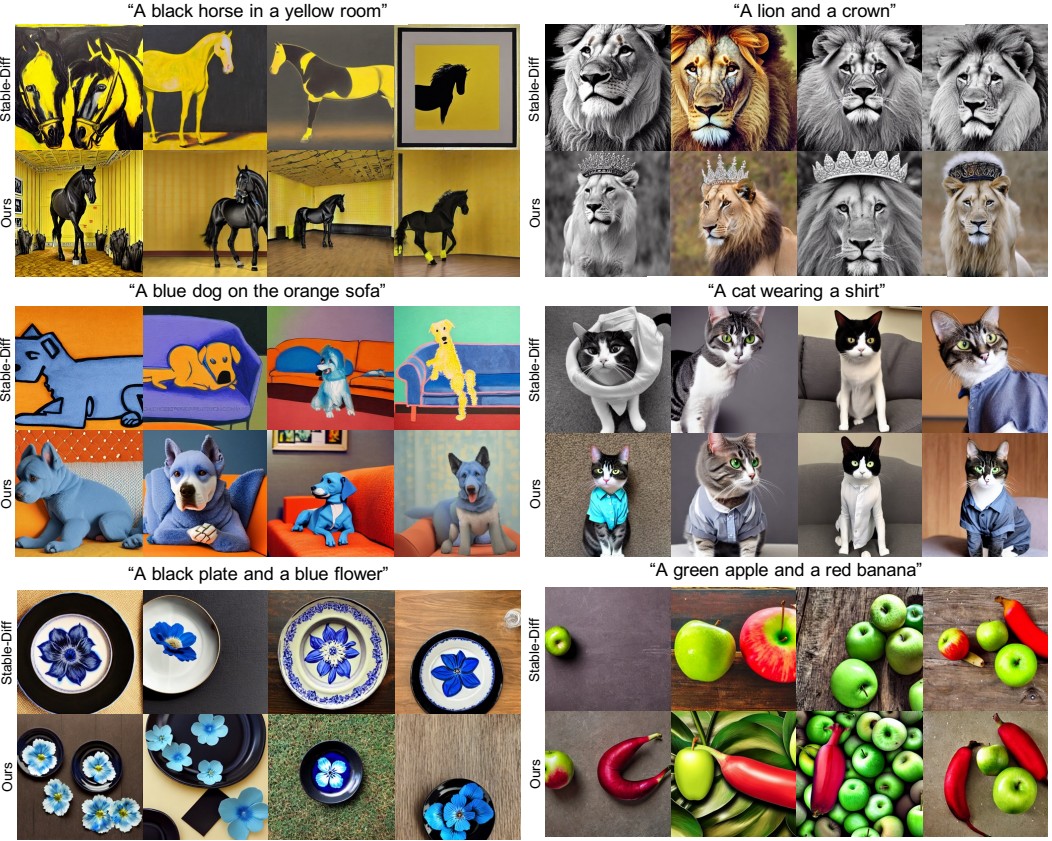

Figure 15: Further results for multi-concept image generation. Best views are displayed.

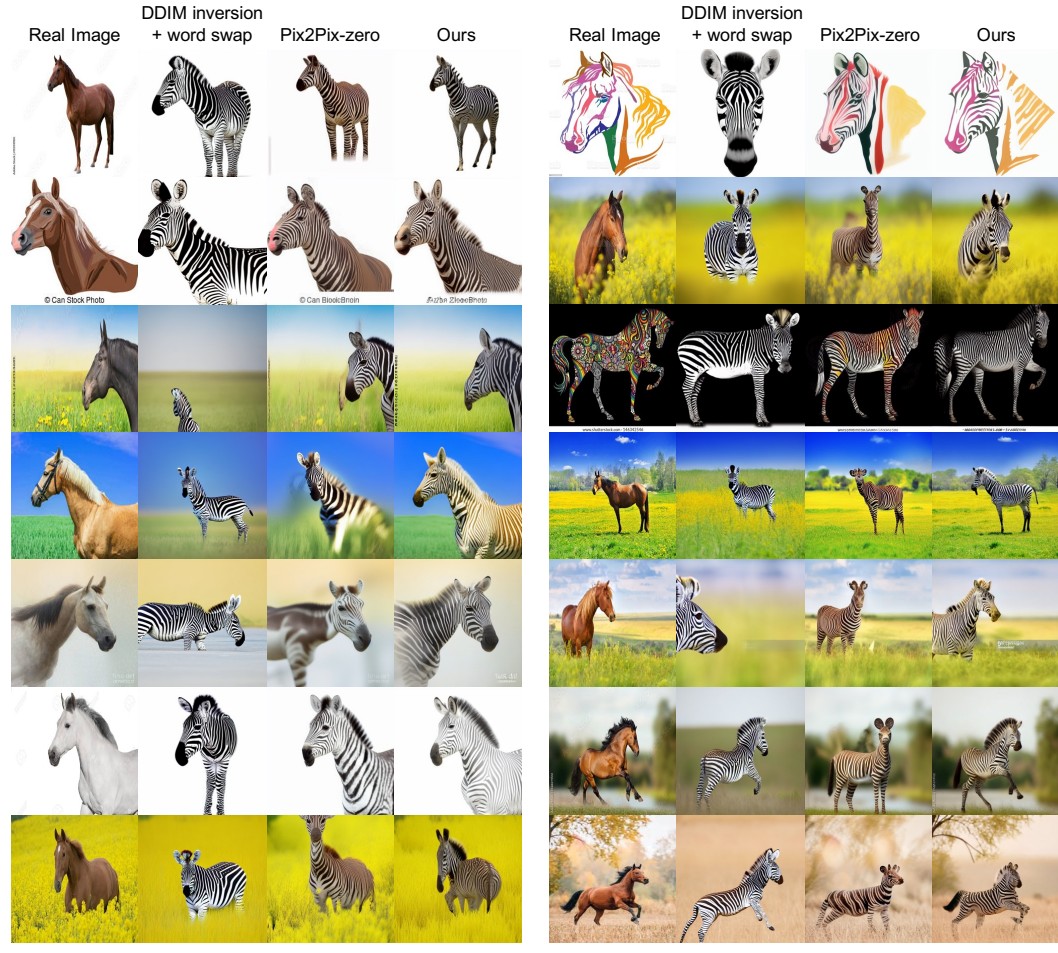

Figure 16: Further results for real image editing: horse to zebra.

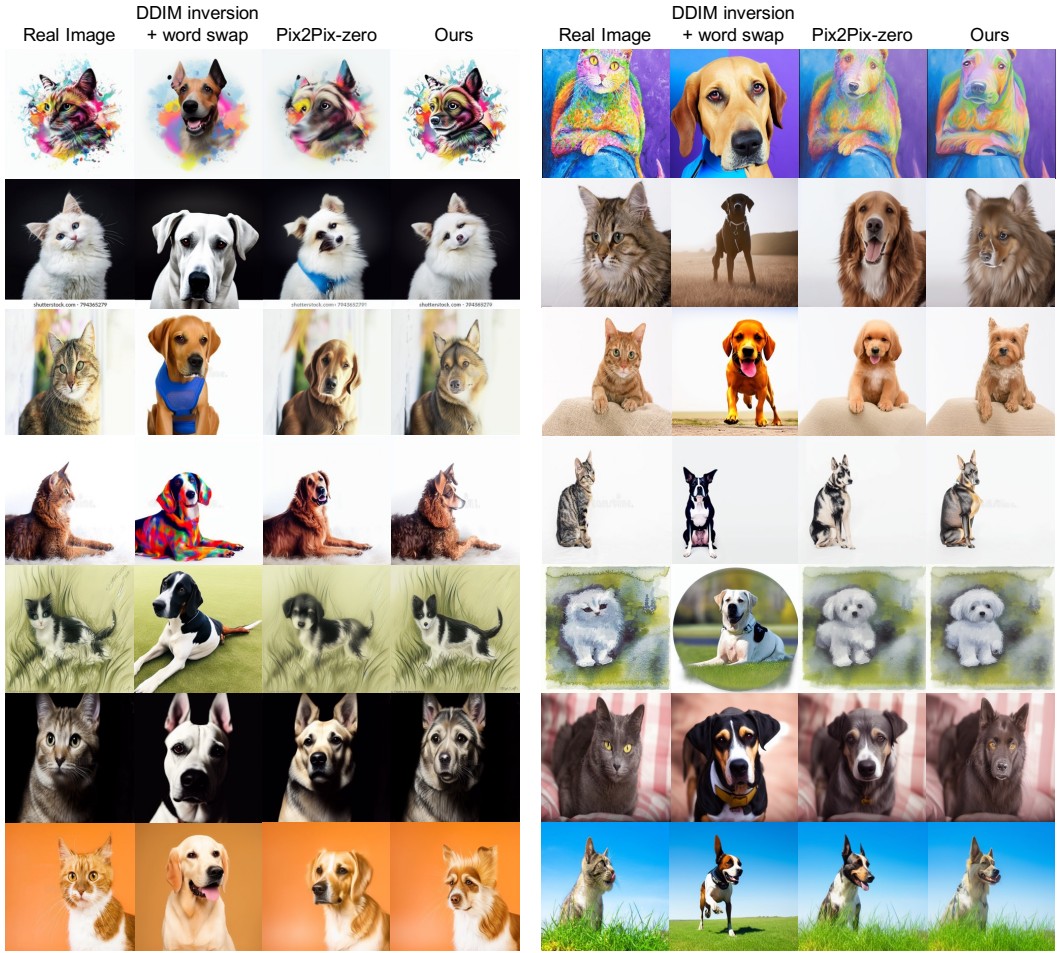

Figure 17: Further results for real image editing: cat to dog.

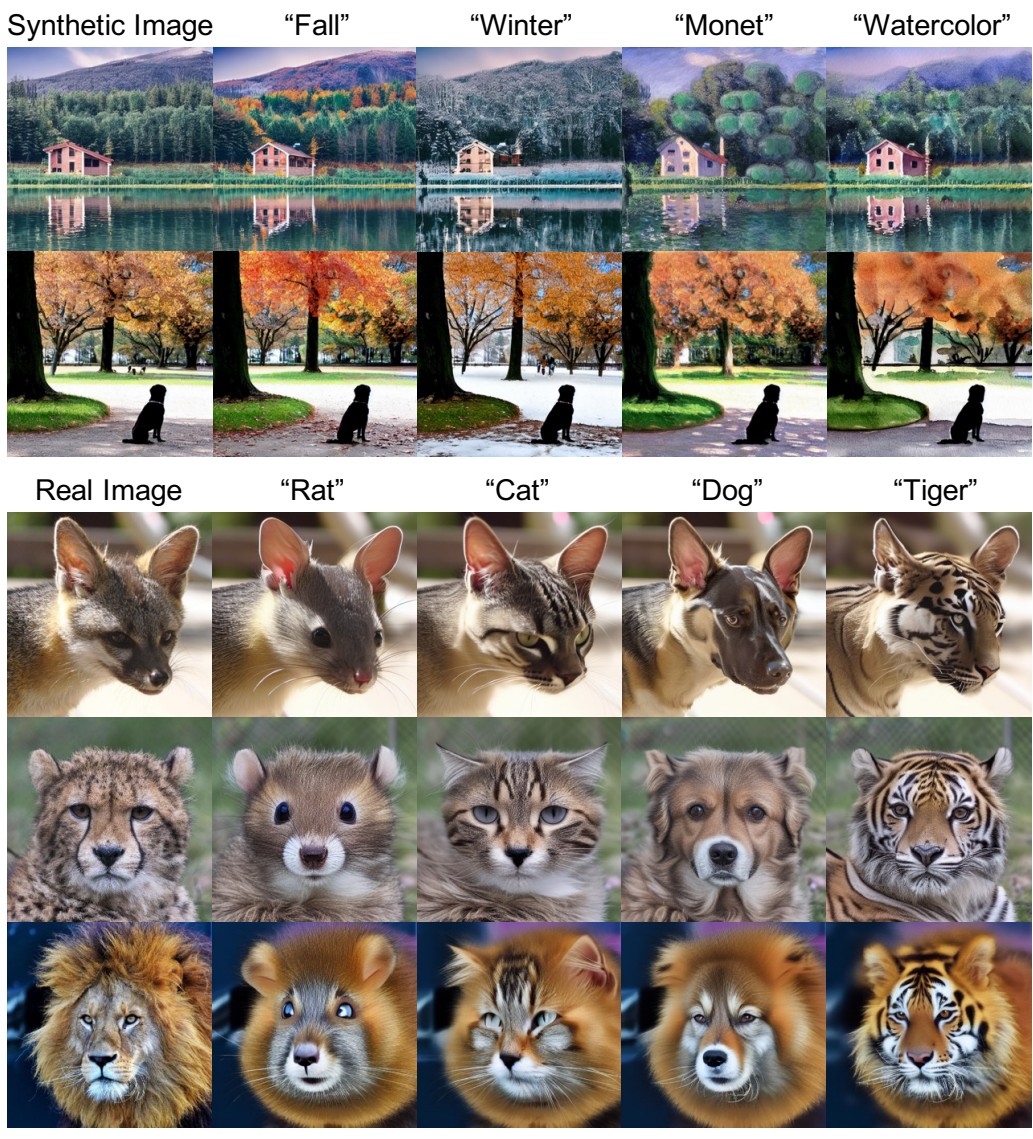

Figure 18: Further results for image editing with varying text prompts. Best views are displayed.

