# Appendix: Energy-Based Cross Attention for Bayesian Context Update in Text-to-Image Diffusion Models

## A Proof of Theorem 1

**Theorem 1.** *For the energy functions*

$$\mathrm{E}(\boldsymbol{Q}; \boldsymbol{K}) = \frac{\alpha}{2} \operatorname{diag}(\boldsymbol{K}\boldsymbol{K}^T) - \sum_{i=1}^{N} \operatorname{logsumexp}(\boldsymbol{Q}\boldsymbol{k}_i^T, \beta) \tag{17}$$

*and*

$$E(\boldsymbol{K}) = \log \sum_{i=1}^{N} \exp(\frac{1}{2}\boldsymbol{k}_i\boldsymbol{k}_i^T), \tag{18}$$

*the gradient of the log posterior is given by:*

$$\nabla_{\boldsymbol{K}} \log p(\boldsymbol{K} \mid \boldsymbol{Q}) = \operatorname{softmax}_2\left(\beta\boldsymbol{K}\boldsymbol{Q}^T\right)\boldsymbol{Q} - \left(\alpha\boldsymbol{I} + \boldsymbol{D}\left(\operatorname{softmax}\left(\frac{1}{2}\operatorname{diag}(\boldsymbol{K}\boldsymbol{K}^T)\right)\right)\right)\boldsymbol{K}, \tag{19}$$

*Then, by using the chain rule the update rule of context vectors $\boldsymbol{C}$ is derived as follows:*

$$\boldsymbol{C}_{n+1} = \boldsymbol{C}_n + \gamma\left(\operatorname{softmax}_2\left(\beta\boldsymbol{K}\boldsymbol{Q}^T\right)\boldsymbol{Q} - \left(\alpha\boldsymbol{I} + \boldsymbol{D}\left(\operatorname{softmax}\left(\frac{1}{2}\operatorname{diag}(\boldsymbol{K}\boldsymbol{K}^T)\right)\right)\right)\boldsymbol{K}\right)\boldsymbol{W}_K^T, \tag{20}$$

*where $\gamma > 0$ is a step size, and $\boldsymbol{D}(\cdot)$ is a vector-to-diagonal-matrix operator.*

*Proof.* Based on the Bayes' theorem, the gradient of the log posterior is derived as:

$$\nabla_{\boldsymbol{K}} \log p(\boldsymbol{K} \mid \boldsymbol{Q}) = -\left(\nabla_{\boldsymbol{K}} \mathrm{E}(\boldsymbol{Q}; \boldsymbol{K}) + \nabla_{\boldsymbol{K}} \mathrm{E}(\boldsymbol{K})\right). \tag{21}$$

First, with definition (17),

$$\nabla_{\boldsymbol{K}} \mathrm{E}(\boldsymbol{Q}; \boldsymbol{K}) = \alpha\boldsymbol{K} - \nabla_{\boldsymbol{K}} \sum_{i=1}^{N} \operatorname{logsumexp}(\boldsymbol{Q}\boldsymbol{k}_i^T, \beta), \tag{22}$$

where $\forall i \in \{1, \ldots, N\}$,

$$\begin{aligned}
\nabla_{\boldsymbol{k}_i} \sum_{i=1}^{N} \operatorname{logsumexp}(\boldsymbol{Q}\boldsymbol{k}_i^T, \beta) &= \frac{1}{\beta} \nabla_{\boldsymbol{k}_i} \log \sum_{j=1}^{P_l^2} \exp(\beta\boldsymbol{q}_j\boldsymbol{k}_i^T) \\
&= \sum_{j=1}^{P_l^2} \frac{\exp(\beta\boldsymbol{q}_j\boldsymbol{k}_i^T)}{\sum_{n=1}^{P_l^2} \exp(\beta\boldsymbol{q}_n\boldsymbol{k}_i^T)} \boldsymbol{q}_j \\
&= \operatorname{softmax}(\boldsymbol{Q}\boldsymbol{k}_i^T)^T\boldsymbol{Q}.
\end{aligned} \tag{23}$$

Then, by considering that $\boldsymbol{k}_i$ is a $i$-th row vector of $\boldsymbol{K}$,

$$\begin{aligned}
\nabla_{\boldsymbol{K}} \sum_{i=1}^{N} \operatorname{logsumexp}(\boldsymbol{Q}\boldsymbol{k}_i^T, \beta) &= \left(\operatorname{softmax}_1(\beta\boldsymbol{Q}\boldsymbol{K}^T)\right)^T\boldsymbol{Q} \\
&= \operatorname{softmax}_2(\beta\boldsymbol{K}\boldsymbol{Q}^T)\boldsymbol{Q},
\end{aligned} \tag{24}$$

where the last equality holds due to the definition of $\mathrm{softmax}_1$ in Section 2.2.

Second, with definition (18), $\nabla_{\boldsymbol{K}} \mathrm{E}(\boldsymbol{K}) = \nabla_{\boldsymbol{K}} \log \sum_{i=1}^{N} \exp(\frac{1}{2}\boldsymbol{k}_i\boldsymbol{k}_i^T)$, where

$$
\begin{aligned}
\nabla_{\boldsymbol{k}_i} \log \sum_{i=1}^{N} \exp(\frac{1}{2}\boldsymbol{k}_i\boldsymbol{k}_i^T) &= \frac{\exp(\frac{1}{2}\boldsymbol{k}_i\boldsymbol{k}_i^T)}{\sum_{j=1}^{N}\exp(\frac{1}{2}\boldsymbol{k}_j\boldsymbol{k}_j^T)}\boldsymbol{k}_i \\
&= \mathrm{softmax}\left(\frac{1}{2}\mathrm{diag}(\boldsymbol{K}\boldsymbol{K}^T)\right)_i \boldsymbol{k}_i,
\end{aligned}
\tag{25}
$$

where $\mathrm{softmax}(\cdot)_i$ denotes $i$-th value of a softmax vector. Then,

$$
\nabla_{\boldsymbol{K}} \log \sum_{i=1}^{N} \exp(\frac{1}{2}\boldsymbol{k}_i\boldsymbol{k}_i^T) = \boldsymbol{D}\left(\mathrm{softmax}\left(\frac{1}{2}\mathrm{diag}(\boldsymbol{K}\boldsymbol{K}^T)\right)\right)\boldsymbol{K},
\tag{26}
$$

where $\boldsymbol{D}(\cdot)$ is a vector-to-diagonal-matrix operator that takes $N$-dimensional $\mathrm{softmax}$ vector as an input and returns a $N \times N$ diagonal matrix with $\mathrm{softmax}$ values as main diagonal entries. Then, By combining (22), (24) and (26), one can finally obtain:

$$
\nabla_{\boldsymbol{K}} \log p(\boldsymbol{K} \mid \boldsymbol{Q}) = \mathrm{softmax}_2\left(\beta\boldsymbol{K}\boldsymbol{Q}^T\right)\boldsymbol{Q} - \left(\alpha\boldsymbol{I} + \boldsymbol{D}\left(\mathrm{softmax}\left(\frac{1}{2}\mathrm{diag}(\boldsymbol{K}\boldsymbol{K}^T)\right)\right)\right)\boldsymbol{K}.
\tag{27}
$$

By using the chain rule with $\boldsymbol{K} = \boldsymbol{C}\boldsymbol{W}_K$, the update rule of context vectors $\boldsymbol{C}$ is derived as in (20). $\qquad\square$

We introduce vector-to-matrix operator $\boldsymbol{D}(\cdot)$ to avoid confusion and fix the typo in the main paper.

## B  Pseudo-code for BCU and CACAO

This section provides the description of the pseudocode for the proposed Bayesian Context Update (BCU) and Compositional Averaging of Cross-Attention Output (CACAO). Algorithm 1 outlines the cascaded context propagation across cross-attention layers within the UNet model during the sampling step $t$. Note that the context is reinitialized at the beginning of each sampling step. On the other hand, Algorithm 2 details the BCU implemented in each cross-attention layer. Remark that $\boldsymbol{D}$ in line 5 denotes vector-to-diagonal-matrix operator. Specifically, the proposed BCU provides a significant computational efficiency by reusing the similarity $\boldsymbol{Q}\boldsymbol{K}^T$, which requires computational cost $\mathcal{O}(N^2)$, to compute $\nabla_{\boldsymbol{K}} E(\boldsymbol{Q}; \boldsymbol{K})$. Consequently, there is only a small amount of additional computational overhead associated with the proposed BCU.

---

**Algorithm 1** Context cascade at sampling step $t$

---

**Require:** $\boldsymbol{Q}_t, \boldsymbol{C}_{clip}$, UNet
1: $\boldsymbol{C}_t \leftarrow \boldsymbol{C}_{clip}$ // Re-initialize
2: **for** layer in UNet **do**
3:     **if** layer is CrossAttention **then**
4:         $\boldsymbol{Q}_t, \boldsymbol{C}_t \leftarrow \mathrm{layer}(\boldsymbol{Q}_t, \boldsymbol{C}_t)$ // Algorithm 2
5:     **else**
6:         $\boldsymbol{Q}_t \leftarrow \mathrm{layer}(\boldsymbol{Q}_t)$
7:     **end if**
8: **end for**
9: $\boldsymbol{Q}_{t+1} \leftarrow \boldsymbol{Q}_t$
10: **return** $\boldsymbol{Q}_{t+1}$

---

**Algorithm 2** Bayesian Context Update (BCU)

---

**Require:** $\boldsymbol{Q}, \boldsymbol{C}, \boldsymbol{W}_q, \boldsymbol{W}_k, \boldsymbol{W}_v, \alpha, \beta, \gamma_{\text{attn}}, \gamma_{\text{reg}}$
1: $\boldsymbol{Q}, \boldsymbol{K}, \boldsymbol{V} \leftarrow \boldsymbol{Q}\boldsymbol{W}_q, \boldsymbol{C}\boldsymbol{W}_k, \boldsymbol{C}\boldsymbol{W}_v$
2: $\boldsymbol{S} = \boldsymbol{Q}\boldsymbol{K}^T$
3: $\boldsymbol{Q} \leftarrow \text{softmax}_2(\beta\boldsymbol{S})\boldsymbol{V}$
4: $\nabla_{\boldsymbol{K}} E(\boldsymbol{Q}; \boldsymbol{K}) = \text{softmax}_2(\beta\boldsymbol{S}^T)\boldsymbol{Q}$
5: $\nabla_{\boldsymbol{K}} E(\boldsymbol{K}) = -(\alpha\boldsymbol{I} + \boldsymbol{D}(\text{softmax}(\frac{1}{2}\text{diag}(\boldsymbol{K}\boldsymbol{K}^T))))\boldsymbol{K}$
6: $\Delta\boldsymbol{C} = (\gamma_{\text{attn}}\nabla_{\boldsymbol{K}} E(\boldsymbol{Q}; \boldsymbol{K}) + \gamma_{\text{reg}}\nabla_{\boldsymbol{K}} E(\boldsymbol{K}))\boldsymbol{W}_k^T$
7: $\boldsymbol{C} \leftarrow \boldsymbol{C} + \Delta\boldsymbol{C}$
8: **return** $\boldsymbol{Q}, \boldsymbol{C}$

---

Algorithm 3 outlines the pseudocode for the CACAO implemented for $M$ given contexts. For the simplicity, we exclude the BCU from the algorithm. Nontheless, the BCU and the CACAO could be leveraged together.

---

**Algorithm 3** Compositional Averaging of Cross-Attention Output (CACAO)

---