# OpenReview forum: "Energy-Based Cross Attention for Bayesian Context Update in Text-to-Image Diffusion Models"
_NeurIPS.cc/2023/Conference — NeurIPS 2023 poster_

### Official Review · Reviewer_tGPJ · 2023-07-06

**Soundness:** 3 good
**Presentation:** 3 good
**Contribution:** 3 good
**Rating:** 5
**Confidence:** 2

**Summary:**

This paper proposed a training-free algorithm to modife cross-attention during inference time that could implicitly minimize the energy in the latent space. This paper formulate this idea from a energy perspective. The authors conduct three experiments - multi-concept generation, image inpainting, compositional editing to demonstrate the effectivenns of the new idea.

**Strengths:**

1. The idea from energy perspective is innovative and the derivation of the equations are solid.
2. Figure 2 an the associate caption is straightforward that shows the relation between energy value and multi-concept generation quality.

**Weaknesses:**

1. In experiment section, the authors provide only some generation cases for subjective-quality analysis without any quantitative results. The shown cases after cherry-pick maybe not adequate to demonstrate the effectiveness of the method proposed in this paper.
2. A minor typo in Line 83: "update rule for a state pattern \epsilon" -> "update rule for a state pattern \zeta"

**Questions:**

1. You metions "nested hierarchy of energy functions" many times? But what is the definition of it and there's any reference papers?
2. In Eq. (6), what is the definition of softmax_1 and softmax_2? And how to obtain Eq. (6) from Q=softmax_2(\betaQK^T)K? I am a little confused.

---

> ### Author Rebuttal · Authors · 2023-08-09
>
> **W1: Lack of quantitative results**. We would like to kindly remind the reviewer that the quantitative comparison results have been reported in appendix D. We will move the results to the main paper in our revised version to emphasize the effectiveness of the proposed method. Also, per requests from other reviewers, we additionally evaluate the quality of generated images using Lpips, which again reveal that the proposed method outperforms baseline methods. Please refer to the general comment 2.
>
> **W2: typo**. We will fix it in the revised paper.
>
> **Q1: nested hierarchy of energy functions**. We have used this term to emphasize that there are many energy functions for each layer, time-step, etc. We do not optimize just a single energy but optimize a nested hierarchy of energy functions during the forward pass of updated context vectors. Specifically, we mentioned “nested hierarchy of energy functions” in order to indicate a unified model-specific energy for the entire UNet. We gently note that it is non-trivial to derive an analytical form of the unified energy for a whole U-Net network, mainly due to the inherent complexities arising from non-linearities.
>
> **Q2: definition of softmax_1 and softmax_2, derivation of Eq. (6)**. The definition for subscripts below the softmax is described in lines 64-68, section 2.2.
> eq. (6) could be derived from eq. (4). Eq. (6) is opted for drawing a connection between transformer attention and hopfield energy minimization. For a such connection, the value matrix $V$ is arbitrarily introduced with a mapping $W_V$. For more details, please refer to eq.(10) in [1].
>
> **References**
>
> [1] Ramsauer, Hubert, et al. "Hopfield networks is all you need." arXiv preprint arXiv:2008.02217 (2020).

---

> > ### Comment · Reviewer_tGPJ · 2023-08-19
> > **Thanks for your rebuttal**
> >
> > Dear authors,
> >
> > Thanks very much for your comprehensive rebuttal and discussion for all reviewers. I have carefully read all the discussion and gone through the paper once more. I understand it better, and I think it will be helpful for the image generation community.
> >
> > However, I didn't derive all the equations by myself and read all the reference papers, so I can only give a low confidence (2) and keep my original score (5). Please revise the organization or presentation to make the paper more understandable, as other reviewers suggested.

---

> > > ### Author Response · Authors · 2023-08-20
> > > **Thanks for the positive response from the reviewer**
> > >
> > > We sincerely appreciate the positive feedback and support for our work. It is particularly heartening to hear that our work will be helpful for the image generation community. Moreover, we are pleased that our rebuttal and discussions help the reviewer's understanding. We are certain your insights, feedback, and suggestions have improved our work. Rest assured, we will revise the main paper to incorporate sufficient materials and discussions from this rebuttal period, while also enhancing its comprehensibility. Thank you for the positive feedback and constructive discussion again.

---

### Official Review · Reviewer_SVoL · 2023-07-06

**Soundness:** 2 fair
**Presentation:** 2 fair
**Contribution:** 2 fair
**Rating:** 3
**Confidence:** 4

**Summary:**

This paper proposes a novel **energy-based** framework that can automatically **update the context** used in cross-attention **without additional training**. They claim the proposed updating process well solves the **semantic misalignment** issue in text-to-image diffusion models.

**Strengths:**

1. The paper exhibits a clear, engaging, and concise writing style.
2. The paper introduces an interesting energy definition and proves the subtle connections with the attention mechanism used in transformers.
3. The experimental results look promising.

**Weaknesses:**

1. The definition of the energy function does not appear reasonable. For instance, consider the definition of E(K) in equation 8. E(K) increases as the L2-Norm of k_i, where i = 1, 2, ..., N, increases. The minimum value of E(K) is achieved when all k_i converge to zero. This suggests that the author assumes k_i = 0 has the highest probability (since energy is minimum), which contradicts reality. Additionally, the definition of E(Q;K) in equation 7 also seems unconventional. It resembles E(K;Q) rather than E(Q;K). I have significant doubts regarding the effectiveness of the proposed energy definition. I kindly request the authors to provide a more explicit explanation for why they have defined the energy function in such a manner and clarify the actual significance of these proposed energy functions.

2. There is a lack of quantitative experimental results. While the authors present some promising outcomes, there is a scarcity of extensive quantitative experiments to demonstrate that the proposed framework statistically outperforms the naive cross-attention mechanism. It is imperative to include detailed information about the quantitative experiments conducted to validate the efficacy of the proposed framework.

3. There is no explicit explanation of why energy minimization can help solve semantic misalignment. I can not draw direct connections between them in the paper.

I tend to reject this paper.

**Questions:**

1. The definition of the energy function does not appear reasonable.
2. Lack of quantitative experimental results.
3. Lack of explanations of why energy minimization help slove semantic misalignment.

---

> ### Author Rebuttal · Authors · 2023-08-09
>
> In contrast to your concerns, it appears that there are several misunderstandings by the reviewers. To clarify the misunderstanding, we would like to give detailed point-by-point answers below.
>
> **W1 and Q1: The definition of the energy function does not appear reasonable**. Thanks for the important comment. We would like to gently note that the design of proposed posterior energy $E_{posterior}(K; Q) = E_{likelihood}(Q;K) + E_{prior}(K)$ is based on the theoretical foundations of [1] and empirical analyses in figure 9, 10 (appendix).
>
> **Hopfield energy**. Ramsauer et al. [1] propose a modern Hopfield energy $E_{hopfield}(Q; K) = \frac{1}{2} \text{diag}(Q^TQ) - \sum \text{logsumexp}(Kq_i^T)$ of which query update rule is equivalent to the attention mechanism. It includes two different terms: (a) exponential interaction term $\text{logsumexp}$, and (b) the regularization term $\frac{1}{2} \text{diag}(Q^TQ)$ which ensures that the norm of the queries remains finite and the energy is appropriately bounded.
>
> **Design of E(Q; K) and E(K)**. While $E_{hopfield}(Q; K)$ is proposed for a query update, for a context update, we inherit the principles of [1] and define our likelihood energy as $E_{likelihood}(Q; K) = \frac{\alpha}{2} \text{diag}(K^TK) - \sum \text{logsumexp}(Qk_i^T)$, which is symmetric to the $E_{hopfield}(Q; K)$.
>
> That being said, as Reviewer suggested, employing $E_{likelihood}(Q; K)$ in eq(7) as $E_{posterior}(K; Q)$ appears to be a viable option. However, we have observed inferior performance, and we suspected that the problem stems from the design of regularization term.
>
> Specifically, $\frac{\alpha}{2} \text{diag}(K^TK)$ uniformly regularizes the context vector which often over-penalizes contexts and causes the context to vanish. To resolve this issue, instead of uniformly regularizing every contexts, we set $\alpha=0$ and propose $E_{prior}(K) = \text{logsumexp}(\frac{1}{2} diag(KK^T))$ to regularize the smooth maximum of context vectors. By summing $E_{likelihood}(Q; K)$ and $E_{prior}(K)$, we finally get the posterior energy $E(K; Q)$ to be minimized by Bayesian Context Update (BCU), Eq. (12).
>
> Ablation studies in Fig 9 and 10 support our intuitions. Specifically, $\gamma_{reg}=0$ in Fig 9 suggests that the norm regularization is necessary for the quality and semantic alignment of generated images. In other words, regularizer plays a role in preventing a single context vector from excessively dominating the forward attention path. Moreover, Fig 10 suggests that increasing $\alpha$ may over-penalize the contexts which prohibit using $E_{likelihood}(Q; K)$ in eq(7) as $E_{posterior}(K; Q)$.
>
> **W2 and Q2: Lack of quantitative experimental results**. In contrast to your comment, we would like to note that the quantitative comparison results have already been reported in appendix D. We will move the results to the main paper in our revised version to emphasize the effectiveness of the proposed method. Also, per requests from other reviewers, we additionally evaluate the quality of generated images using Lpips, which again reveal that the proposed method outperforms baseline methods. Please refer to the general comment 2.
>
> **W3 and Q3: Lack of explanations of why energy minimization help slove semantic misalignment**. Please refer to the general comment 1. We would like to clarify that the goal of the proposed method is to achieve the *adaptive* context propagation through UNet, which could mitigate the semantic misalignment caused by the use of fixed context embeddings. Here, we would like to remark that minimizing the energy function is equivalent to maximizing the log-likelihood of the probability density function. In this context, minimizing the energy function $E(K|Q)$ corresponds to maximizing the log-likelihood of $p(K|Q)$, which indicates finding the most likely K given Q and finally achieves the *adaptive* context for a better semantic alignment.
>
> **References**
>
> [1] Ramsauer, Hubert, et al. "Hopfield networks is all you need." arXiv preprint arXiv:2008.02217 (2020).

---

> > ### Comment · Reviewer_SVoL · 2023-08-16
> > **Thank you for your rebuttal**
> >
> > The rebuttal solves part of my issues, especially the experimental part. I am excited to see that the author managed to simply manipulate the attention operation via manually defined energy to achieve better generation results without any fine-tuning or relying on any prior. I am willing to accept the paper if all the experiments are reliable. Although, for me, the theory is not that reasonable or at least not smooth or natural. For instance, the author didn't reply about the actual meaning of why they defined $E(K)$ in this form. They only define $E(K)$ to penalize the norm of $Q$.  And that seems to be the only reason why they call their method `Bayesian`, which is a kind of patchwork.
> >
> > Considering that the other reviewers all agree to accept the paper, I will upgrade my rate to borderline accept

---

> > > ### Author Response · Authors · 2023-08-17
> > > **Thanks for the positive response from the reviewer**
> > >
> > > > Considering that the other reviewers all agree to accept the paper, I will upgrade my rate to borderline accept.
> > >
> > > Thank you very much for your constructive feedback and the decision to adjust the score to *“borderline accept”*. We are truly encouraged by your enthusiasm towards our improved results. We are certain your insights and the discussion have improved our work.
> > >
> > > Furthermore, we kindly wish to bring to your attention that the updated score has **not** yet been reflected in the review. We would be grateful if you could kindly update this as mentioned. Thank you for your consideration.
> > >
> > > > I am willing to accept the paper if all the experiments are reliable. Although, for me, the theory is not that reasonable or at least not smooth or natural. For instance, the author didn't reply about the actual meaning of why they defined $E(K)$ in this form. They only define $E(K)$ to penalize the norm of $Q$. And that seems to be the only reason why they call their method Bayesian, which is kind of patchwork.
> > >
> > > Thanks for your positive response.  We will make sure that all the experimental results will be incorporated into the final version of the paper so that the results can be reproduced reliably.
> > >
> > > Additionally, we wish to emphasize to the reviewer that introducing the prior $E(K)$  is not a patchwork, nor is its intent to drive all $k_i$ to zero. Rather, its main objective is to prevent any element from escalating to unjustifiably high magnitudes.
> > >
> > > This is evident when considering the properties of the log-sum-exponential function. Notably, the log-sum-exponential is recognized as a smooth (i.e. differentiable) approximation to the “max” operation, as illustrated by the inequality:
> > >
> > > $\max(x_1,\cdots, x_N) \leq \log \sum_{i=1}^N x_i \leq \max(x_1,\cdots, x_N) +\log N$
> > >
> > > Thus, by penalizing the max value with the prior term, we aim to constrain its elements from escalating to unrealistic magnitudes. Figures 8, 9, and 10 in our work corroborate these insights. Within our Bayesian framework, this approach offers a relaxed yet plausible prior. The underlying presumption is that a realistic value ought to be finite, and this prior simply represents one facet of the knowledge we wish to apply.
> > >
> > > Addressing your initial comment --"the minimum value of E(K) is achieved when all k_i converge to zero. This suggests that the author assumes k_i = 0 has the highest probability ... which contradicts reality"-- it's essential to note that penalizing the maximum value does not inherently push its values to zero. This is particularly true when the prior is paired with the likelihood $E(Q;K)$.
> > >
> > > Historical precedence in statistical literature supports this perspective. Take, for instance, Rissanen's renowned “universal prior for integers” used for the model estimation [1]. While it does penalize the model order $N$ through $\log_*()$ function, Bayesian model estimations employing this “universal prior for intergers” don't drive the estimated model order to zero. The popularity of the universal prior for model estimation stems from its relaxed and realistic approach; it asserts a finite realistic model order, representing just one facet of prior knowledge. This mirrors the principles of our prior model.
> > >
> > > ---
> > >
> > > **Reference**
> > >
> > > [1] Rissanen, Jorma. "A universal prior for integers and estimation by minimum description length." The Annals of statistics 11.2 (1983): 416-431.

---

> > > ### Author Response · Authors · 2023-08-20
> > > **Thanks and a Kind Reminder**
> > >
> > > Dear Reviewer  SVoL,
> > >
> > > Thank you for acknowledging our contribution and your decision to increase the score to "Borderline Accept (5)".
> > >
> > > However, as the deadline for the discussion period nears, we've noticed that the suggested score update hasn't been reflected.  In case you forget or are not aware, we would like to kindly remind the reviewer that you can edit the score in the initial review to reflect your suggested change.
> > >
> > > Your prompt attention and feedback on this matter would be greatly appreciated.

---

### Official Review · Reviewer_5Nym · 2023-07-07

**Soundness:** 4 excellent
**Presentation:** 3 good
**Contribution:** 3 good
**Rating:** 6
**Confidence:** 3

**Summary:**

The paper proposes to formulate cross-attention layers using energy-based models such that by minimizing the cross attention energy with respect to the context latent representation, the method can further alleviate semantic misalignments between generated or edited samples and the input descriptions, and allow zero-shot compositional generalization using combinations of cross attention outputs based on multiple inputs.

**Strengths:**

1. The paper introduces a detailed theoretical formulation of cross attentions using energy-based perspective.
2. The method outperforms existing methods qualitatively on various image generation tasks (e.g., inpainting, multi-concept generation and compositional generation) without additional training.
3. The authors have conducted relatively comprehensive evaluations on the method.

**Weaknesses:**

1. **Lack of limitations**. We have seen the good side of such method, which is to improve semantic alignment between generated images and text inputs. However, there are often trade-offs. For example, energy-based optimization is quite unstable, so it requires some tuning in terms of step size and the number of optimization steps. It would be good to mention the limitation of the method, for example, how much the method is slower compared to the standard denoising steps.
2. **Lack of experiments on Image quality**. Though I think only context vectors are optimized, image generation quality will be unlikely to change much. I would still suggest evaluating generated images would provide a way to help understand whether such algorithms can change the generation performance.  In addition, although semantic alignment seems to improve in qualitative results, we also need to measure it quantitatively. And current metrics (e.g., CLIP) for measuring semantic alignment can be unreliable, thus a human evaluation seems necessary to compare different methods quantitatively.

**Questions:**

1. is the method quite sensitive to the hyper-parameters that you use to optimize context vectors for each image? Based on the supplementary material, it seems to me that the method is quite sensitive and often requires a lot of tuning in terms of coefficients $\gamma_{attn}, \gamma_{reg}$ for updates, step size, etc.

**Limitations:**

The authors have included limitations.

---

> ### Author Rebuttal · Authors · 2023-08-09
>
> **W1: Lack of limitations**. Thanks for the comment. We only modified the forward path of the cross-attention layer and this is equivalent to a one-step gradient descent of defined energy. Although we did not minimize the energy multiple steps to achieve further convergence, we have shown that the one-step energy update (i.e. one-step forward step) effectively reduces the energy, as shown in figure 2, and it results in better performance on multiple tasks. In addition, for the modified forward path, we reuse the $QK^T$ term (algorithm 2 in appendix B), which significantly reduces the additional computational cost. In fact, both Stable Diffusion and the proposed method take ~7.0 seconds to generate a single image within 50 sampling steps with NVIDIA GeForce RTX 3090. However, we acknowledge the reviewer's comment that the proposed method requires step size as hyperparameters, namely $\gamma_{attn}$ and $\gamma_{reg}$, which the user should tune for their specific applications. We address this issue in appendix D by providing guidance on the range of appropriate values for these parameters.
>
> **W2: Lack of experiments on Image quality**. Per the reviewer’s request, we also measured LPIPS to evaluate the quality of the generated image and conducted human evaluation. Please refer to the general comment 2.
>
> **Q1: Sensitivity to the hyper-parameter**. We would like to remark that the hyperparameters, namely $\gamma_{attn}$ and $\gamma_{reg}$, correspond to step size for one-step gradient descent of energy which is natural to tune to get the better performance. However, we also would like to emphasize that fixed hyperparameters are used in the real-image editing task. Remarkably, the proposed method outperforms the state-of-the-art works (see appendix D for the detailed result). This result indicates that the proposed method exhibits robustness within a certain range of hyperparameters, which may vary depending on the task. In order to facilitate the use of the proposed method, we have included a guide for selecting hyperparameters for each task in appendix C. Still, there is a need for tuning $\gamma$ for a newly suggested task so we will add it to the limitation section.

---

> > ### Comment · Reviewer_5Nym · 2023-08-18
> >
> > Thanks the authors for the rebuttal and it has resolved some of my questions.
> >
> > I will keep my rating as it is.

---

> > > ### Author Response · Authors · 2023-08-18
> > > **Thanks for the positive response from the reviewer**
> > >
> > > We sincerely appreciate the positive feedback and support for our work with maintaining a high rating. It is particularly heartening to hear that our rebuttal addressed the reviewer's concerns in a positive direction. We are certain your insights and the discussion have improved our work. Rest assured, we will revise the main paper to incorporate sufficient materials. Thank you for the positive feedback and constructive discussion again.

---

### Official Review · Reviewer_2y1g · 2023-07-13

**Soundness:** 3 good
**Presentation:** 4 excellent
**Contribution:** 4 excellent
**Rating:** 6
**Confidence:** 4

**Summary:**

This work tackles the semantic misalignment problem of stable diffusion model using the energy-based model framework.The authors first show that each cross-attention in the diffusion model can be seen as one step optimization of a pre-defined energy function. They then formulate a Bayesian update for the context vector accordingly. The authors demonstrate the effectiveness of their method by showing qualitative samples in multi-concept generation, text-guided image inpainting as well as compositional generation tasks.

**Strengths:**

I love the novelty and theoretical framework of this paper. While there are several other works tackle the more controlable image generation/editing problem based on the modification of the cross attention layers, this paper provides a systematical theory framework for this. This might not only help with the improvement on certain task but also facilicate people to form a deeper understanding of the model itself.

**Weaknesses:**

1. Lack of quantitative results:
While several qualitative samples are shown in the paper, quantitative evaluation over a large number of different samples can be more convincing to judge the model's performance against the baseline. The authors may consider using pretrained model (for example like [1] does) or using human evaluation.

2.Choice of hyper-parameter:
 Looking at the supplementary, it seems that a successful editing requires one to choose parameters $\alpha$, $\beta$, $\gamma$ case-by-case. Furthermore, it seems that changing hyper-parameters have a great influence over the generation performance.  This might limit the proposed method to be used in real applications.

[1] Training Diffusion Models with Reinforcement Learning

**Questions:**

1. The understanding of the whole unet: While we may look at each individual cross-attention layer as updating a certain energy function, I would like to hear the authors' insight on how we can understand the unet as a whole. At different cross-attention layers, the weight are different. And there are a batch of linear and non-linear transformations between each two cross-attention layers. Then can we understand the whole unet as updating a certrain energy? (The cascading update of C seems to suggest that the optimization across the whole unet has some consistency.)  And if possible, how?

2. The energy distribution across different attention layers: In figure 2, the authors plot the energy of 16 attention layers across different sampling steps. It seems that at all the steps, the energy will peak at the middle layer and reach bottom at the two ends. Can the authors provide some insight on why this happens?

3. In section 3.1 algorithm, the authors provide their design of updating context vector C. While this seems to work well on the samples shown in the paper, I'm wondering what if other update algorithms are use. For example, what if we don't cascade $C_{n+1, t}$ to the next (n+1)th layer but updates $C$ for all the layers starting from scratch and update them for more than one step?

4. From figure 8 in the supplementary, it seems that increasing $\gamma_{reg}$ instead of $\gamma_{attn}$ encourages pattern to occur.  When $\gamma_{reg}$ is zero, no matter how to tune $\gamma_{attn}$, the editing cannot get successful results (no teddy bear here). This can be a bit counter-intuitive. As for my understanding, the attention term should be the one to align the key and query. Are there any explanations on why these happen?



**Limitations:**

 The authors have adequately addressed the limitations.

---

> ### Author Rebuttal · Authors · 2023-08-09
>
> **W1: Lack of quantitative results**. We would like to kindly remind the reviewer that we have already measured CLIP accuracy and DINO-ViT structure distance motivated by [1,2] for the image editing task and compared it with state-of-the-art methods in the appendix. The result shows that the proposed method exhibits the best editing performance while preserving the structure of the original input image. Please refer to the appendix for details.  We will move the quantitative results to the main paper in our revised version.
>
> Furthermore, per other reviewers' suggestions, we also conducted a human evaluation and measured LPIPS to evaluate image quality. See the general comment 2.
>
> **W2: Choice of hyper-parameter**.
>
> - $\alpha$: We would like to assure the reviewer that we found $\alpha$ to be the most stable and effective when set to 0, as we mentioned in lines 146-147 and the ablation study results in figure 10 in the supplementary. Therefore, we used it consistently in all experiments, eliminating the need for separate tuning.
>
> - $\beta$: In the case of $\beta$, it plays a role as the temperature of the softmax in the conventional attention operation. Thus, following the lead of previous attention-related studies, we highly recommend setting $\beta$ to $\frac{1}{\sqrt{d}}$, where $d$ denotes the dimensionality of the embeddings.
>
> - $\gamma$: For the hyperparameters, called $\gamma_{attn}$ and $\gamma_{reg}$, we would like to remark that they correspond to step size for one-step gradient descent of energy which is natural to tune to get the better performance. However, we also would like to emphasize that fixed hyperparameters are used in the real-image editing tasks, instead of case-by-case tuning. Remarkably, the proposed method outperforms the state-of-the-art works (see appendix D for the detailed result). This result indicates that the proposed method exhibits robustness within a certain range of hyperparameters. In order to facilitate the use of the proposed method, we have included a guide for selecting hyperparameters for each task in appendix C.
>
> **Q1: The understanding of the whole unet**. Thanks for the constructive question. The paper aims to interpret each individual cross-attention layer from the energy perspective, as the reviewer points out. Formulating a unified energy for the entire UNet architecture is non-trivial, mainly due to the inherent complexities arising from non-linearities. Although it would be out of the scope of this work, we would like to recommend a related work [3, 4] that tries to understand the whole Transformer from an energy perspective.
>
> **Q2: The energy distribution across different attention layers**. Thanks for carefully reading the manuscript. As the reviewer mentioned, there is a tendency for energy along cross-attention layers in Unet. It may be related to the dimensions of feature maps which have not been considered for plotting the energy dynamics (for example, the energy is highest at the bottleneck layer of Unet). However, we would like to note that the key message conveyed by Figure 2 is the relative energy gap before and after applying BCU. The figure highlights the impact of the proposed method on energy reduction. Furthermore, it is evident that the energy difference increases significantly, implying that adaptive context propagation influences the energy of subsequent CA layers, leading to cumulative energy minimization. To ensure clarity and emphasis on the relative energy gap, we will revise Figure 2 in the main paper.
>
> **Q3: Using other update algorithms**. Per the reviewer’s request, we implement a multistep context update for comparison while keeping the propagation of C to subsequent layers. For the multi-concept generation and inpainting tasks, we observe that this multiple update of the context vector actually improves the quality of generated images. This is in the same line as our perspective that one forward path of context vector is equivalent to one-step gradient descent for energy function. Please refer to Figure 3 in the rebuttal pdf for the result. That being said, we also observe that the multiple context update is relatively computationally expensive and a single-step update is usually sufficient for improved performance. Therefore, we decide to use a single-step update with context propagation to subsequent layers.
>
> **Q4: $\gamma_{reg}$ and $\gamma_{attn}$ in Figure 8**. The observation is consistent with Eq. (12), where the regularization term, controlled by $\gamma_{reg}$, plays a role in preventing a single context vector from excessively dominating the forward attention path (lines 127-129). In other words, without proper regularization, other context vectors could dominate the attention path, leading to the neglect of the ‘teddy bear’ concept. We would like to remark that the Stable-Diffusion receives 77 tokens to generate the image. However, with the proper application of the regularization term (step size 0.025 for figure 8 in the supplementary), we can observe that the BCU with non-zero $\gamma_{attn}$ can correctly generate the teddy bear. This result highlights the significance of the proposed BCU in capturing and expressing concepts given by textual conditions but also emphasizes that both the attention term and the regularization term need to be considered together.
>
> **References**
>
> [1] Tumanyan, Narek, et al. "Plug-and-play diffusion features for text-driven image-to-image translation." Proceedings of the IEEE/CVF Conference on Computer Vision and Pattern Recognition. 2023.
>
> [2] Parmar, Gaurav, et al. "Zero-shot image-to-image translation." ACM SIGGRAPH 2023 Conference Proceedings. 2023.
>
> [3] Yang, Yongyi, and David P. Wipf. "Transformers from an optimization perspective." Advances in Neural Information Processing Systems 35 (2022): 36958-36971.
>
> [4] Hoover, Benjamin, et al. "Energy transformer." arXiv preprint arXiv:2302.07253 (2023).

---

> > ### Comment · Reviewer_2y1g · 2023-08-14
> >
> > I extend my gratitude to the authors for their commendable work and thoughtful rebuttal. Their responses have addressed my initial concerns, inclining me towards endorsing this paper. However, as other reviewers have raised additional critiques that I find valid, I will maintain my "weak acceptance" rating for now and await their feedback on whether their concerns have been sufficiently addressed.

---

> > > ### Author Response · Authors · 2023-08-14
> > > **Thanks for the positive response from the reviewer**
> > >
> > > We sincerely appreciate the positive feedback and support for our work. It is particularly reassuring to note that our rebuttal effectively addressed the reviewer's concerns, especially regarding the lack of quantitative results and sensitivity of hyper-parameters. Again, we extend our thanks to the reviewer for the careful and constructive review.

---

### Official Review · Reviewer_jCTY · 2023-07-18

**Soundness:** 2 fair
**Presentation:** 2 fair
**Contribution:** 3 good
**Rating:** 5
**Confidence:** 3

**Summary:**

This paper proposes an energy-based model (EBM) framework addresses semantic misalignment in text-to-image diffusion models by incorporating EBMs in each cross-attention layer, minimizing a nested hierarchy of energy functions, and achieving highly effective results in diverse image generation tasks. From the shown figures, they demonstrate the effectiveness of the EBM framework.

**Strengths:**

1, The proposed framework demonstrates general applicability to diverse tasks, such as inpainting, image editing, and multi-concept generation.

2, Theoretical analysis and explanations are comprehensively conducted to enhance the robustness of the experimental results.

3, Notably, this paper stands out as the first diffusion model study observed to incorporate an energy-based perspective into its formulation.

**Weaknesses:**

1, The experimental results are only including the qualitative comparison with some other methods. In my view, some quantitative numbers should also be shown in the paper to demonstrate the performance of this method compared to others.

2, For multi-concept generation, the Custom Diffusion [1] has been released during the end of last year, it should be considered as a comparison.

3, The same comparison limitations also happened in inpainting (considering Blended Diffusion[2], GLIDE[3]) and image editing (Prompt2Prompt[4], pix2pix-zero[5]).

4, For evaluation metrics, there are also commonly used metrics in different sub-area. For example, the CLIP-score/Lpips/Structure-Dist can be considered in image editing tasks.


[1] Multi-Concept Customization of Text-to-Image Diffusion
[2] Blended Diffusion for Text-driven Editing of Natural Images
[3] GLIDE: Towards photorealistic image generation and editing with text-guided diffusion model
[4] Prompt-to-Prompt Image Editing with Cross Attention Control
[5] Zero-shot Image-to-Image Translation

**Questions:**

As I stated above, the main drawback of this paper is in the experimental part. There are lack of qualitative comparisons with correct methods. Instead, the original Stable Diffusion model is always serving as the baseline comparison. Furthermore, there are no sufficient quantitative results to show the efficiency of the proposed framework in this paper.

**Limitations:**

As I stated in the weakness and questions.

---

> ### Author Rebuttal · Authors · 2023-08-09
>
> **W1: Some quantitative numbers should also be shown in the paper**. In contrast to your comment, we would like to remind the reviewer that the quantitative comparison results have already been reported in appendix D. We will move the results to the main paper in our revised version to emphasize the effectiveness of the proposed method. Also, per requests from other reviewers, we additionally evaluate the quality of generated images using Lpips, which again reveal that the proposed method outperforms baseline methods. Please refer to the general comment 2.
>
> **W2: more baseline in multi-concept generation**. Thank you for the suggestions. However, Custom Diffusion is for the personalized generation, which requires a few images that contain the desired concepts. In contrast, the proposed method only depends on a given textual condition to generate multiple concepts simultaneously. Thus, it seems not a fair comparison. Instead, we compare the performance with Composable-diffusion [1] which is commonly compared for the multi-concept generation. Please refer figure 8 in the rebuttal pdf. We will add this result to the revised paper.
>
> **W3: more baseline in inpainting and image editing**. Per your suggestion, we conduct further comparison studies with Blended  Latent Diffusion [2]. Please refer to figure 9 in the rebuttal pdf. We will add the results in the revised paper. Also, we would like to kindly remark that the performance of recent algorithms for image editing already has been compared and reported in appendix D.
>
> **W4: evaluation metrics**. We have used CLIP-acc and Structure-Dist by following [3, 4] (see appendix D). We will move the quantitative comparison results to the main paper in our revised version to emphasize the effectiveness of the proposed method. Also, per the requests from reviewers, we further evaluated the quality of generated images from the real-image editing task via Lpips. Please refer to the general comment 2.
>
> **References**
>
> [1] Liu, Nan, et al. "Compositional visual generation with composable diffusion models." European Conference on Computer Vision. Cham: Springer Nature Switzerland, 2022.
>
> [2] Avrahami, Omri, Ohad Fried, and Dani Lischinski. "Blended latent diffusion." ACM Transactions on Graphics (TOG) 42.4 (2023): 1-11.
>
> [3] Tumanyan, Narek, et al. "Plug-and-play diffusion features for text-driven image-to-image translation." Proceedings of the IEEE/CVF Conference on Computer Vision and Pattern Recognition. 2023.
>
> [4] Parmar, Gaurav, et al. "Zero-shot image-to-image translation." ACM SIGGRAPH 2023 Conference Proceedings. 2023.

---

> > ### Comment · Reviewer_jCTY · 2023-08-14
> > **Thanks for your response**
> >
> > Thanks to your rebuttal reply. I already proceeded to review the supplementary material to check your referenced information. Consequently, I would like to address my final inquiry: in the context of multi-concept composition, are you achieving something similar to Attend-and-Excite [1] framework? Is Attend-and-Excite comparable to your proposed methodology?
> >
> > [1] attend-and-excite: attention-based semantic guidance for text-to-image diffusion models

---

> > > ### Author Response · Authors · 2023-08-14
> > > **Thanks for the additional comment.**
> > >
> > > We appreciate the reviewer's follow-up query and are pleased to provide further clarification.
> > >
> > > In response to the reviewer's query, our answer is affirmative. In general, *Attend-and-Excite* performs well for multi-concept as it is designed for that specific goal. However, the proposed method would be better for textual conditions beyond the simple union of concepts (e.g. A and B, A with B). This is largely attributed to the effectiveness of the adaptive context update.
> > >
> > > The *Attend-and-Excite* is motivated by the intuition that "at least one patch in its attention map should exhibit a high activation value for a token to be manifested in the generated image". Consequently, *Attend-and-Excite* updates latent at each time point to ensure that the most neglected subject token is more attended in the cross-attention layer. In our context, this could be classified as a query-only-update mechanism with a fixed context.
> > >
> > > While *Attend-and-Excite* effectively generates images based on the textual condition that contains a union of two concepts, it might be limited for more complex textual conditions. For example, we observe that *Attend-and-Excite* is prone to ignore the relationship between two concepts (e.g. A cat *wearing* a shirt, A blue dog *on* the orange sofa).
> > >
> > > In contrast, the proposed method adaptively updates the context including each concept and the relationship. In fact, the proposed method successfully generates images that reflect not only multiple concepts but also their relationship (please refer to figure 11 in appendix).
> > >
> > > Unfortunately, we're constrained from uploading supplementary figures or links at this time.
> > > However, we would like to suggest that the reviewer tries out the *Attend-and-Excite* demo on HuggingFace.
> > > - prompt 'A cat wearing a shirt' with seed [0] and token_indices [2,5]
> > > - prompt 'A blue dog on the orange sofa' with seed [0] and token_indices [2,3,6,7]
> > >
> > > Note that the result is similar even if we apply the *Attend-and-Excite* to the relationship tokens (e.g. wearing, on).

---

> > > > ### Comment · Reviewer_jCTY · 2023-08-14
> > > > **Thanks for your response again**
> > > >
> > > > Thanks for your reply to answer my concerns.
> > > >
> > > > Actually I'm more convinced with the main paper + the supplementary material + the rebuttal files. I am happy to improve the score to "Borderline accept". The reason that I cannot make it even higher is that there is no opportunity to modify the paper in the rebuttal stage. If so, I would wonder how good if the authors can reorganize the paper content from all these sufficient materials. From the main paper only, it is still having some weakness. But overall, the method deserves an acceptance.

---

> > > > > ### Author Response · Authors · 2023-08-14
> > > > > **Thanks for the positive feedback**
> > > > >
> > > > > We sincerely appreciate the positive feedback and raising the score. It is particularly heartening to hear that our rebuttal effectively addressed the reviewer's concerns. We believe that the discussion with the reviewers has further improved our work, and we are grateful for the constructive feedback. We agree that it would be beneficial to incorporate the quantitative results and additional results from this rebuttal period into the main paper. Rest assured, we will revise the main paper to incorporate sufficient materials.

---

> > > > > > ### Comment · Reviewer_jCTY · 2023-08-17
> > > > > > **Thanks for your response again**
> > > > > >
> > > > > > Thank you for your continued efforts in addressing my inquiries. I am revisiting your paper and have a question aimed at gaining a clearer understanding of its content.
> > > > > >
> > > > > > Given that your proposed EBM based modifications involve the cross-attention mechanism to reduce energy within the latent space, I'm curious whether you've conducted any visualizations illustrating how the cross-attention maps evolve, particularly in the context of real images during tasks such as inpainting or real image editing. There you can compare with the DDIM inversion cross-attention maps.
> > > > > >
> > > > > > As I understand that attaching images within this response might not be feasible, a comparison through numerical data or descriptive depictions would be greatly appreciated. Sorry for the inconvenience. I'm just confusing and curious.

---

> > > > > > > ### Author Response · Authors · 2023-08-18
> > > > > > > **Thanks for your constructive suggestion**
> > > > > > >
> > > > > > > **Experiment setting.** Thanks for the constructive question and positive response. Your suggestion serves to enhance our paper's clarity, and we welcome the opportunity to provide a comprehensive response. Per your suggestion, we present a comprehensive comparison between the cross-attention maps produced by the DDIM inversion technique and the proposed method within the context of real-image editing. We select a representative image featuring a white cat$\rightarrow$dog, located in Figure 13, row 5. Subsequently, we embark on the visualization of cross-attention maps, focusing on a designated ```dog``` token, during the image generation with (**a**) DDIM inversion or (**b**) the proposed method. In consonance with your suggestion, we chart the evolutionary trajectory of cross-attention maps through sequential sampling steps, each spaced at intervals of 10 steps (0 steps, 10 steps, 20 steps, ...).
> > > > > > >
> > > > > > > **Results.** Upon the initial phase (0 steps), no substantial distinction is evident between the attention maps, as the attention values in both maps are quite randomly distributed. This similarity arises from the initial absence of semantic cues related to the depicted dog within the spatial query.
> > > > > > >
> > > > > > > However, as the process unfolds, marked differences emerge. Specifically, our attention map progressively converges upon key discriminative features distinguishing between cat and dog, such as the face, ears, etc, with minimal emphasis on the structural attributes of the source image, such as background and posture. In contrast, the DDIM inversion's attention map undergoes a distinct transformation. It progressively shifts focus away from the original source image's overarching structure, and instead converges on the newly generated dog subject. This shift entails modifications in positioning, posture, identity, or gaze orientation compared to the source image. We note that these differences between attention maps begin to emerge at early steps, i.e. 10 steps.
> > > > > > >
> > > > > > > These findings suggest that the proposed method effectively maintains the overarching structure of spatial query representations while also adaptively reflecting the semantically encoded contextual information to the designated query representations. It is noteworthy that the control of step size ($\gamma$) and a residual path for context update (Eq. (12)) may avoid the catastrophic devastation of the fundamental structure of the source image. While we're unfortunately constrained from uploading supplementary figures or links at this moment, we will certainly incorporate these new results into the revised paper. Again, we extend our thanks to the reviewer for the careful and constructive suggestion.

---

### Official Review · Reviewer_MvGR · 2023-07-28

**Soundness:** 2 fair
**Presentation:** 2 fair
**Contribution:** 2 fair
**Rating:** 5
**Confidence:** 3

**Summary:**

This paper proposes  Bayesian Context Update (BCU) and  Compositional Averaging of Cross-Attention Output (CACAO). The main idea is to view the cross-attention between text and image as optimizing an energy-based model, and modify the intermediate outputs according to the energy. A series of examples are shown to demonstrate the effectiveness of BCU and CACAO.

**Strengths:**

1. The examples are visually intriguing.
2. The energy viewpoint of the cross-attention mechanism in Stable Diffusion could be enlightening.

**Weaknesses:**

1. I can understand that Q_t is the H * W * C dimensional feature in the network processing the latent-space image at time step t, and C_t is the context vector which is an optimizable variable added by the authors. C_t is initialized to C_clip, then it is optimized going through the layers. As the layers go deeper, C_t is shifted from C_clip, which means the semantic meaning is shifted from the original user text input. My question is: how severe is the shift? How do the authors control the shift? In fact, it makes more sense to me if Q_t,l are the variables to be optimized.

2. More clarification on the examples in Figure 2 is required. I tried Stable Diffusion by myself, and found Stable Diffusion can generate "A monkey and a hat" with the correct semantic meaning. I kindly ask the authors to elaborate how the Stable Diffusion examples are picked. Moreover, Stable Diffusion + Ours degrades the image quality significantly (all the images become black-and-white). Could the authors provide some explanation on why that happens?

3. More quantitative results should be added to prove the effectiveness of the proposed method. In the provided qualitative examples, the advantage of the proposed method over Composable-Diff and Plug-and-Play is marginal. Since the results of generative models are heavily dependent on the random seeds, I suggest the authors to provide systematic quantitative evaluation to benchmark the proposed method. I do notice the quantitative evaluation part in the Appendix, but only animal transition seems limited.

**Questions:**

Please refer to Weakness.

**Limitations:**

Limitations are properly discussed at the end of the paper.

---

> ### Author Rebuttal · Authors · 2023-08-09
>
> **W1: How severe is the shift? How do the authors control the shift?**. Thanks for pointing out the important question that is deeply related to the motivation of our work. We would like to emphasize that the change of context vector is adaptive, not an invalid shift, because the proposed BCU allows an adaptive propagation of the context vector through UNet, resulting in a context vector that is better aligned with $Q_t$ on cross-attention space. For more details, please refer to the general comment 1. Moreover, we would like to remark that control of step size ($\gamma_{reg}$) and a residual path for context update (Eq. (12)) can avoid deviating outside the appropriate semantics.
>
> **Experiment.** To further support our claim, we use the updated $C_T$ ($T$ for a number of sampling time steps) from the proposed method as a fixed context vector instead of $C_{clip}$ and perform multi-concept generation using the conventional cross-attention operation. Since the energy functions are defined differently for each sampling time step, using $C_T$ as a fixed context for each sampling time step may result in low-quality samples. However, this approach allows us to evaluate whether the updated context contains the correct semantics of the given textual conditions, in contrast to $C_{clip}$. Figure 2 in the rebuttal pdf demonstrates that the updated context vector does indeed capture the correct semantics of the given textual conditions (e.g., both black horse and yellow room). We will include this result in the revised paper to further strengthen the evidence for the effectiveness of the proposed method.
>
> **W2: More clarification on the examples in Figure 2, degradation of the image quality**. We would like to note that when the Stable Diffusion can generate correct semantic meaning, the improvement via the proposed method is marginal (refer to figure 7 in the rebuttal pdf). Therefore, we sampled multiple times with random seeds from 0 to 40, and improve the performance of the incorrect Stable diffusion sample cases.
>
> For Figure 2 in the main paper, we recognize that increasing $\gamma_{attn}$ is helpful to avoid gray image generation.  Please refer to Figure 6 in the rebuttal pdf. This is related to the role of $\gamma_{reg}$ which is a regularization term. Finally, we would like to kindly note that our method does not significantly degrade the image quality. Please compare the 1st and 3rd rows of Figure 3. Indeed, the curated black-and-white images are the outcomes inherent to the behavior of the original stable diffusion model upon which our approach is founded.
>
> **W3: More quantitative results**.  Per the requests from reviewers, we further compare the performance of human face editing and measure additional metrics such as LPIPS and human evaluation to evaluate the quality of generated images. Furthermore, we compare the proposed method with additional baseline methods. Please refer to the general comment 2. Lastly, we would like to remark that the provided quantitative evaluations are by following the prior works [1, 2] on this research area.
>
> **References**
>
> [1] Tumanyan, Narek, et al. "Plug-and-play diffusion features for text-driven image-to-image translation." Proceedings of the IEEE/CVF Conference on Computer Vision and Pattern Recognition. 2023.
>
> [2] Parmar, Gaurav, et al. "Zero-shot image-to-image translation." ACM SIGGRAPH 2023 Conference Proceedings. 2023.

---

> > ### Comment · Reviewer_MvGR · 2023-08-14
> > **Thank you**
> >
> > The authors conducted additional experiments and clarified the unclear parts. Therefore I increased my score to 5. I suggest the authors to incorporate the new results in the later versions of their paper. Thank you for the efforts!

---

> > > ### Author Response · Authors · 2023-08-14
> > > **Thanks for the positive response from the reviewer**
> > >
> > > We sincerely appreciate you taking the time to provide your insightful feedback and raising the score. We are certain your insights and the discussion have improved our work. We will incorporate your suggestion and new results into the next version of our paper.

---

### Official Review · Reviewer_nFGp · 2023-07-28

**Soundness:** 2 fair
**Presentation:** 2 fair
**Contribution:** 2 fair
**Rating:** 5
**Confidence:** 4

**Summary:**

This paper aims to optimize the context representation through energy based formulation of the cross-attention within the U-Net at test-time to achieve semantic alignment between the textual representation and the image features in the U-Net.
Experiments are performed for the multi-concept generation, text-guided inpainting and image editing.

**Strengths:**

+ The motivation of the paper is clear and is supported with adequate experiments.
+ The approach allows for compositional generation. The qualitative results show that the approach does better than simple DDIM+inversion.
+ Multiple concepts can be composed together in the generated images with test-time optimization.


**Weaknesses:**

- The formulation of the gradient posterior in Eq. 9 is missing the term on the expectation of the gradient on the right side?
- The difference and the motivation to optimize the energy wrt the keys instead of queries is not so clear? Ablations with the two approaches can be done to show the benefits and attention visualizations may help to evaluate the difference between the two.
- The subscripts 1, 2 below the softmax are not defined.
- In lines 150-151 what is meant by the "forward path of cross-attention"?
- Low sample diversity: Example in figure 2 seems to harm the sample diversity wrt Stable diffusion.
- What is the computational overhead? How many optimization steps are required to converge to a good solution?


**Questions:**

Please see weaknesses above.

Specifically following clarifications are required:

1. The motivation for optimizing keys over queries.
2. The role of the regularization term in Eq. 12.
3. In theorem 1 the updates are wrt to the keys, however, eq 14 considers the energy wrt to the queries. The whole idea  of the paper in terms of novelty  from theorem 1, then why use eq. 14?

The results and comparisons with the baselines can be moved to the main paper. Theorem 1 and Eq 14 need to be better discussed (better combined in one section) and highlighting the contributions and the prior work clearly.

**Limitations:**

Limitations are adequately discussed.

---

> ### Author Rebuttal · Authors · 2023-08-09
>
> **W1: The formulation of the gradient posterior in Eq. 9**. We would like to clarify that the notation E in eq. (9) denotes the energy function defined in eq. (7) and (8), not the expectation. We realize a typo in eq. (1) that should be corrected as \mathbb{E}. We will fix eq. (1) in our revised paper.
>
> **W2: The difference and the motivation to optimize the energy wrt the keys instead of queries**. We would like to clarify that optimization for the energy w.r.t the queries have already been done by the cross-attention operation as described in proposition 1 and eq. (6). For the difference and the motivation to optimize the energy w.r.t key, please refer to the general comment 1.
>
> **W3, 4: The subscripts 1, 2 below the softmax, Forward path of cross-attention**. The definition for subscripts below the softmax is described in lines 64-68, section 2.2. The “forward path of cross-attention” means one operation of the conventional cross-attention, which is described in eq. (6). While eq. (6) only outputs updated query Q, the proposed BCU means updating the context vector C according to eq. (12) and propagating it to the subsequent UNet layer. Please refer to figure 1 in the rebuttal pdf file.
>
> **W5: Low sample diversity**. Regarding figure 2 in the main paper, it seems that the influence of $\gamma_{reg}$ is significant, causing the monkeys to appear more grayish because the role of “reg” is to normalize the context. If we were to increase $\gamma_{attn}$ by two-folds, the result would inherit the diversity of the Stable Diffusion (see figure 6 in the rebuttal pdf). We will include a discussion of this in our revised paper. That said, we would like to point out that the proposed method can generate more diverse images compared to Stable-Diffusion that  may not accurately reflect the context and lead to low diversity as shown in figure 3 in the main paper and figure 11 in the appendix. Thus, the proposed method proves to be more beneficial in generating a wider range of images through textual guidance.
>
> **W6: Computational overhead and optimization steps**. We would like to emphasize that the additional computational cost is negligible. Both Stable Diffusion and the proposed method take 7.0 seconds to generate a single image within 50 sampling steps with NVIDIA GeForce RTX 3090. Specifically, for the modified forward path, we reuse the $QK^T$ term (algorithm 2 in appendix B), which significantly reduces the additional computational cost. Also, we would like to clarify that we have not *explicitly* optimized image representations and context embeddings. The primary contribution of the proposed method is to establish a link between cross-attention operation and energy minimization. This allows us to effectively modify the cross-attention operation, resulting in adaptive propagation of the context embedding that better aligns the image features than a fixed CLIP embedding. Throughout our work, we have demonstrated that a single step of cross-attention, which is equivalent to one-step gradient descent for a defined energy function, is sufficient to improve performance for various tasks. In particular, we can observe the emergence of multiple concepts related to given textual conditions in the early phase of generation, which implies a fast convergence of the energy function (figure 2 in the main paper).
>
> **Q1: The motivation for optimizing keys over queries**. Please see the general comment 1.
>
> **Q2: The role of the regularization term**. The regularization term in Eq. 12 originated from the conditional energy function w.r.t K (7) and the role is described in lines 127-132. In summary, it constrains the energy of each context vector $k_i$, preventing it to explode during the maximization of logsumexp($Qk_i$, $\beta$). Specifically, $\gamma_{reg}=0$ in Fig 9 suggests that the norm regularization is necessary for the quality and semantic alignment of generated images. Our regularizer plays a role in preventing a single context vector from excessively dominating the forward attention path. For more details, please see the general comment 1.
>
> **Q3: eq. 14**. As mentioned by the reviewer, the primary contribution of the paper is theorem 1. However, we would like to highlight that the essence of the paper is the energy perspective interpretation of the cross-attention operation. In other words, what we have proposed in the paper are two different forward paths (each for query and key) of cross-attention layers that aim to minimize different energies (eq. 14 and eq. 9). Please refer to figure 1 in the rebuttal pdf.
>
> In this context, we could readily extend the energy perspective interpretation to compositional generation by formulating the task as sampling from the posterior distribution as described in equation (13), leading to the novel energy (14) and method CACAO (16). It is important to note that CACAO is designed to focus on the query update, allowing for the effective incorporation of multiple textual conditions into image generation, while BCU enables adaptive context propagation. The compatibility of BCU and CACAO for compositional image generation is evident, as demonstrated by the ablation study result in the appendix.
>
> In summary, the proposed energy perspective interpretation of cross-attention operation is a powerful tool that could be leveraged from a better understanding of the stable-diffusion model to improved performance for various applications.
>
> **Q4: move results to the main paper, better discuss Theorem 1 and Eq 14**. Agreed. Moving the comparison results to the main paper is beneficial and we will revise the manuscript accordingly. Also, we will consider re-organizing the manuscript per reviewers' suggestion to highlight the contribution and main motivations of this work.

---

> > ### Comment · Reviewer_nFGp · 2023-08-14
> > **Thank you for the rebuttal**
> >
> > I have read the rebuttal and the concerns raised have been adequately addressed. That said, the main limitation of the work is in the organization which the authors promise to revise.  Taking into account that the peer reviewers are positive that these changes will be made and included in the later versions I will raise my score to 5. I urge the authors to also include a discussion in the main paper or in the supplemental on the computational cost for completeness. Thank you.

---

> > > ### Author Response · Authors · 2023-08-15
> > > **Thanks for the positive feedback**
> > >
> > > We truly appreciate the positive feedback and raising the score. We are pleased to hear that our rebuttal could adequately address the reviewer's concerns. We certainly agree that it would be beneficial to incorporate the quantitative results and additional results from this rebuttal period into the main paper. Rest assured, we will revise the main paper to incorporate sufficient materials. Thank you for the constructive discussion again.

---

### Author Rebuttal · Authors · 2023-08-09

We sincerely thank all the Reviewers for their valuable comments. We are encouraged that the reviewers say that “the motivation of the paper is clear” (nFGp), “love the novelty and theoretical framework” (2y1g), “the idea from energy perspective is innovative” (tGPJ) and “Theoretical analysis and explanations are comprehensively conducted” (jCTY), etc.

We have summarized some of the major concerns that were raised from the reviewers below. Point-to-point responses were also included as a reply to each reviewer.

---

**Comment 1**. We further clarify the motivation and contributions of the proposed work in a more concise way. Please refer to the conceptual model figure 1 in the uploaded rebuttal pdf.

**Main motivation**: Our main point is that the conventional use of fixed context embedding is sub-optimal, as illustrated in Figure 1 of the main paper (Stable Diffusion). Such misalignment may be attributed to various causes: error-prone human-designed prompts, limited capacity of pre-trained textual encoders, fundamental gap between CLIP-space and U-Net latent space, etc.  In order to mitigate the misalignment, instead of leveraging fixed contexts, we aim to establish an adaptive context by modeling p(context|representations). Note that this is a significant departure from the previous methods which only model p(representations|context) with frozen context vectors.

**Idea**: To model p(context|representations), we propose an energy-based modeling. Specifically, the energy function in Eq. (3) is closely related to the attention mechanism as shown in [1]. Inspired from these results, we define p(K|Q) in the cross-attention space with the posterior energy $E_{posterior}(K; Q) = E_{likelihood}(Q;K) + E_{prior}(K)$.  Then, one of the primary contributions of the proposed method is to create a correspondence  between the context and respresentation through the cross-attention layers of a diffusion model by minimizing the parameterized energy functions. In contrast to a static and fixed context embeddings, the proposed method enables the adaptive context propagation through an energy minimization such that both Q and C are updated and propagated to subsequent layers.  This test time optimization for context and respresenation alignment through cross attention have never been tried before, which may lay down foundation for further research.

---

**Comment 2**. Reviewers are kindly reminded that, due to the limited space, we reported multiple results in the appendix of the original submission, which includes ***quantitative comparison* (section D.5)**, ablation study on BCU, CACAO (Fig 9), analysis on prior energy $E(K)$ (Fig 10), more visualizations (Fig 11-14), etc.
In the attached rebuttal PDF file, we now add quantitative comparisons on compositional editing tasks with the CelebA-HQ dataset. Specifically, we focus on three image-to-image translation tasks: woman $\rightarrow$ man, woman $\rightarrow$ woman w/ glasses, and woman $\rightarrow$ man w/ glasses. For this, we select source images of women without wearing glasses from CelebA-HQ. To ensure fair comparisons, all methods leverage the same version of Stable Diffusion, sampler, sampling steps, etc.

Resp_Table 1 shows that the proposed energy-based framework achieves a high CLIP-Acc while maintaining low Structure Dist and LPIPS. Note that the proposed method works consistently well on the compositional editing, i.e., woman $\rightarrow$ man w/ glasses. Moreover, we conduct a user study to evaluate the perceptual quality of generated samples. Following the protocol in [2], we ask 21 people to rate the scores on a scale from 1(poor) to 5(excellent) on the following questions:  Q1) Does the output align with the intended semantics of the target text? (Text-match), Q2) Do the generated images appear realistic? (Realism), Q3) Do the outputs preserve the content information from source images? (Content). Resp_Table 2 shows that the proposed method consistently scored high across all questions. We would like to remark that, for the cat-to-dog case, while the DDIM inversion produces outputs in alignment with the given text, as evidenced by the highest score for Q1, it is notably limited by a significant loss of original content, reflected by the lowest score on Q3. This observation aligns with our findings reported in section D.5 and figure 7 in appendix. We will include all the quantitative results in the revised version of our main paper.

---
**[Resp_Table1] CelebA-HQ results**

|CelebA-HQ | Woman→Man | | | Woman→Woman w/ glasses| |  | Woman→Man w/ glasses| | |
|-|-|-|-|-|-|-|-|-|-|
|**Method**| **CLIP Acc (↑)** | **Dist (↓)** | **LPIPS (↓)** | **l CLIP Acc (↑)** | **Dist (↓)** | **LPIPS (↓)** | **l CLIP Acc (↑)** | **Dist (↓)** | **LPIPS (↓)** |
|CycleDiffusion|57.0%|0.156|0.433| l 74.0%|**0.018**|**0.117**| l 63.5%|0.237|0.443|
|Pix2Pix-zero|93.0%|0.043|0.391| l 66.0%|0.024|0.266| l 93.9%|0.052|0.444|
|Ours|**94.0%**|**0.035**|**0.324**| l **99.0%**|0.029|0.292| l **96.0%**|**0.037**|**0.341**|



**[Resp_Table2] User Study**
| |Cat→Dog | | | l Horse→Zebra| | | l Cat→Cat w/ glasses| | |
|-|-|-|-|-|-|-|-|-|-|
|**Method** | **Q1** | **Q2** | **Q3** | **l Q1** | **Q2** | **Q3** | **l Q1** | **Q2** | **Q3**|
|SDEdit|4.15|4.00|4.10 | l 4.00|4.05|4.55| l 4.40|4.40|4.30|
|DDIM-inv|**4.30**|3.85 |2.35| l 4.40|3.50|2.40| l 3.60|3.10|2.10|
|Pix2Pix-zero|3.35|2.80 |2.80| l 3.70|3.05|3.25| l 3.35|3.15|2.80|
|Ours|3.95|**4.00**|**4.40** | l **4.55**|**4.30**|**4.65**| l **4.70**|**4.50**|**4.70**|

---

**References**

[1]: Ramsauer, Hubert, et al. "Hopfield networks is all you need." arXiv preprint arXiv:2008.02217 (2020).

[2]: Kwon, Gihyun, and Jong Chul Ye. "Diffusion-based image translation using disentangled style and content representation." International Conference on Learning Representations. 2023.

---

### Decision · Program_Chairs · 2023-09-21

**Decision:**

Accept (poster)

**Comment:**

This paper proposes a new energy-based framework for text-to-image generation. Extensive experiments are conducted to show that the proposed framework is effective in handling various image generation tasks. In response to the concerns initially raised by all the reviewers in their initial assessments, the rebuttal systematically addressed these issues, including adding additional comparison results, clarification of the motivation and contribution of the paper, and a discussion of computational cost. Following extensive discussions, a consensus was achieved among all seven reviewers, culminating in the decision to accept the paper. This agreement is attributed to the paper's innovation, theoretical justification, and sufficient empirical validation. The AC agrees with all the reviewers and recommends accepting the paper. To further improve the paper quality, the AC urges the authors to revise their paper by taking into account all the suggestions provided by the reviewers, including integrating extra experimental results conducted during the rebuttal phase.